# Per-channel autoregressive linear prediction padding in tiled CNN processing of 2D spatial data

## Abstract

We present linear prediction as a differentiable padding method that has no train-able parameters. For each channel, a stochastic autoregressive linear model is fitted to the data by minimizing its noise terms in the least-squares sense. The data is iteratively padded with conditional expected values of the autoregressive model. We trained the convolutional RVSR super-resolution model from scratch on satellite image data, using different padding methods. The simplest variant of linear prediction padding reduced the mean square super-resolution error by ∼2% at the image edges, compared to zero and replication padding, with a ∼25% increase in inference time. Linear prediction padding better approximated satellite image data and RVSR feature map data. With zero padding, RVSR appeared to use more of its capacity to compensate for the higher approximation error. Cropping the RVSR output by a few pixels reduced the super-resolution error and suppressed the impact of the choice of padding method, favoring fast zero and replication padding.

## 1   Introduction

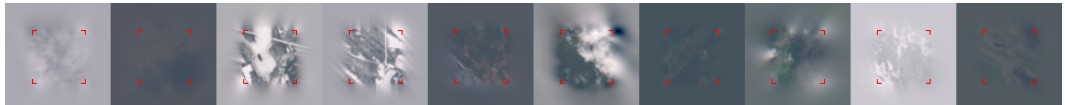

Figure 1: Satellite images padded using our linear prediction padding method (variant `lp6x7`).

Geospatial rasters and other extensive spatial data can be processed in tiles (patches) to work around memory limitations. The results are seamless if all whole-pixel shifts of the tiling grid result in the same stitched results. A convolutional neural network (CNN) consisting of valid convolutions and pointwise operations is equivariant to whole-pixel shifts. In such *shift equivariant* CNN-based processing, each input tile must cover the receptive fields of the output pixels, i.e. input tiles must be overlapped. This wastefully repeats computations. In deep CNNs, the receptive fields may be thousands of pixels wide (Araujo et al. 2019), exacerbating the problem.

Spatial reduction (see Fig. 2) in deep CNNs is commonly compensated for by padding the input of each spatial convolution, with zeros in the case of the typically used *zero padding* (zero for brevity) or with the value of the nearest input pixel in *replication padding* (repl). In the less used polynomial *extrapolation padding* (extr$N$), a Lagrange polynomial of a degree $N-1$ is fitted to the $N$ nearest input pixels from the same row or column, and the padding is sampled from the extrapolated polynomial. extr0 is equivalent to zero and extr1 to repl. Like these methods, our *linear prediction padding* method is channel-wise, stateless and free of trainable parameters. The recent *padding module* (Alrasheedi et al. [2023]) implements stateful multi-channel linear prediction with coefficients trained for prediction of edge pixels of padding input using gradient-based methods.

Submitted to 39th Conference on Neural Information Processing Systems (NeurIPS 2025). Do not distribute.

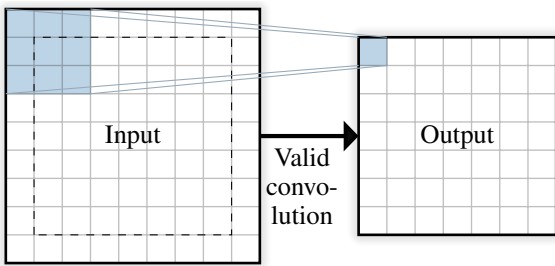

Figure 2: Valid convolution with a $3\times3$ kernel with origin in the middle erodes data spatially by a one-pixel-thick layer at each image edge. The receptive field of a single output pixel (⬜) is illustrated.

An ideal padding method would exactly predict the spatial data outside the padding input view. This is not possible for data coming from an effectively random process, such as natural data or a CNN feature map derived from it. Using padding increases a CNN's error towards tile edges (Huang et al. 2018). Conversely, center cropping the output of a CNN employing padding reduces the error (Huang et al. 2018). In tiled processing, the deviation from shift equivariance can be measured by the mean square deviation between overlapped CNN predictions in neighboring output tiles. The deviation is bounded from above by four times the mean square loss which can therefore be used as a proxy for it (follows from the *parallelogram law*, see appendix A). The strength of the receptive fields of output neurons typically decay super-exponentially with distance to their center (Luo et al. 2016), meaning that an output center crop discarding from each edge fewer pixels than the radius of the theoretical receptive field could reduce the prediction error close to that of a valid-convolution CNN.

We introduce our *linear prediction padding* (`lp`) method in Section 2 presenting variants based on covariance (Section 2.1, variants `lp1x1cs` and `lp2x1cs`, with the numbers denoting the height×width of the prediction neighborhood in vertical padding) and on autocorrelation (Section 2.2, `lp2x1`, `lp2x3` `lp2x5`, `lp3x3`, `lp4x5`, and `lp6x7`). Implementation details are given in Section 2.3. We evaluated the performance of tiled CNN super-resolution employing different padding methods (`lp` variants and `zero`, `repl`, and `extr`$N$) with the super-resolution and evaluation methods given in Section 3, results in section Section 4 and our conclusions and ideas for further work in Sections 5 and 6.

## 2 Linear prediction padding method

Linear prediction is a method for recursively predicting data elements using nearby known or already predicted elements as input (for an introduction, see Makhoul 1975 for the 1D case and Weinlich 2022 for the 2D case, both very approachable texts). Linear prediction is closely related to stochastic autoregressive (AR) processes. We can model a zero-mean version $I = J - \mu$ of 2D single-channel image or feature map data $J$ that has a mean value of $\mu$, by a zero-mean stationary process $\hat{I}$:

$$\hat{I}_{(y,x)+h_0} = \sum_{i=1}^{P} a_i \hat{I}_{(y,x)+h_i} + \varepsilon_{(y,x)+h_0}, \tag{1}$$

where $(y, x)$ are integer spatial coordinates, $a_i$ are coefficients that parameterize the process, $\varepsilon$ is zero-mean independent and identically distributed (IID) noise, and the *extended neighborhood* $h$ is a list of 2D coordinate offsets (relative to a shared origin), first that of the pixel of interest $\hat{I}_{(y,x)+h_0}$ followed by its *neighborhood* $\hat{I}_{(y,x)+h_{1...P}}$ in any order. Each pixel depends linearly on $P$ neighbors (see Fig. 3). Our approach to linear prediction padding is to least-squares (LS) fit the AR model (Eq. 1) to zero-mean data $I$, and then to compute the padding as the expectance of the fitted AR process, assuming for conditional prediction that the data is a realization of the model, $\hat{I} = I$.

The LS fit is obtained by minimizing the mean square prediction error (MSE), with residual noise $\varepsilon$ as error. Assuming real $I$, MSE is calculated from the error by:

$$\varepsilon_{(y,x)+h_0} = I_{(y,x)+h_0} - \sum_{i=1}^{P} a_i I_{(y,x)+h_i}, \quad \text{MSE} = \tfrac{1}{|S|} \sum_{(y,x)\in S} \left( I_{(y,x)+h_0} - \sum_{i=1}^{P} a_i I_{(y,x)+h_i} \right)^2, \tag{2}$$

$$S = \{(y,x) \ : \ (y,x) + h_i \in K \text{ for all } i \in [0, P]\}$$

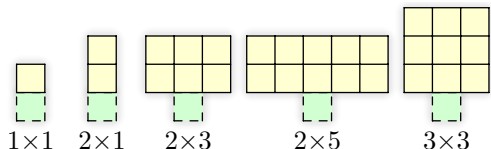

$$1{\times}1 \quad 2{\times}1 \quad 2{\times}3 \quad 2{\times}5 \quad 3{\times}3$$

Figure 3: Illustration of some rectangular 2D neighborhoods (with pixels □) next to the pixel of interest (▨), defining the linear dependency structure of a downwards causal AR model. The extended neighborhood $1{\times}1$ can be defined by $h = [(1,0),(0,0)]$ and $2{\times}1$ by $h = [(2,0),(0,0),(1,0)]$.

where $S$ is the set of coordinates that keeps all pixel accesses in the sums within the set of input image pixel coordinates $K$ (see the left side of Fig. 4).

The noise, being zero-mean by definition, does not contribute to the AR process expected value:

$$\mathrm{E}\left(\hat{I}_{(y,x)+h_0}\right) = \mathrm{E}\left(\sum_{i=1}^{P} a_i \hat{I}_{(y,x)+h_i} + \varepsilon_{(y,x)+h_0}\right) = \sum_{i=1}^{P} a_i \, \mathrm{E}\left(\hat{I}_{(y,x)+h_i}\right) + 0. \tag{3}$$

We can recursively calculate conditional expectances (the padding) further away from known pixels (the input image), using both the known pixels $I_K = \{I_{(x,y)} : (x,y) \in K\}$ and any already calculated expectances (the nascent padding):

$$\mathrm{E}\left(\hat{I}_{(y,x)+h_0} \,\middle|\, I_K\right) = \sum_{i=1}^{P} a_i \begin{cases} I_{(y,x)+h_i} & \text{if } (y,x)+h_i \in K, \\ \mathrm{E}\left(\hat{I}_{(y,x)+h_i} \,\middle|\, I_K\right) & \text{otherwise.} \end{cases} \tag{4}$$

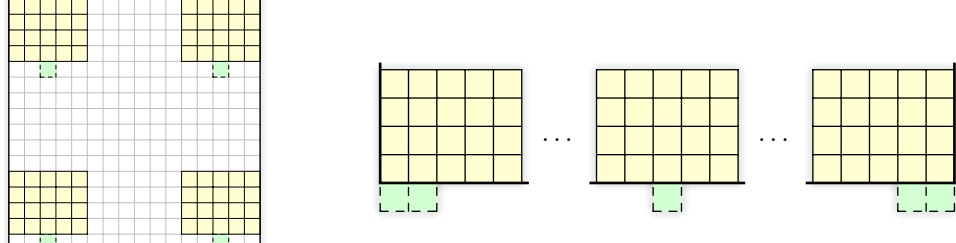

Figure 4: Downwards linear prediction padding with a $4{\times}5$ neighborhood — Left: predicted (▨) and neighborhood pixels (□) at the corners of the rectangular area of coordinates over which MSE is calculated during fitting. Right: corner handling prevents narrowing of the recursive prediction front.

Our linear prediction padding process is illustrated in Fig. 5. We use rotated extended neighborhoods to pad in different directions and adjusted extended neighborhoods (see the right side of Fig. 4) to pad near the corners. We pad channels of multi-channel data separately. Before padding, we make the data zero-mean by mean subtraction, which we found necessary for numerically stable Cholesky solves of AR coefficients and to meet the assumption of a zero-mean AR process. We add the mean back after padding. Depending on the extended neighborhood shape, we use a method based on covariance or a method based on autocorrelation which is faster for larger neighborhoods.

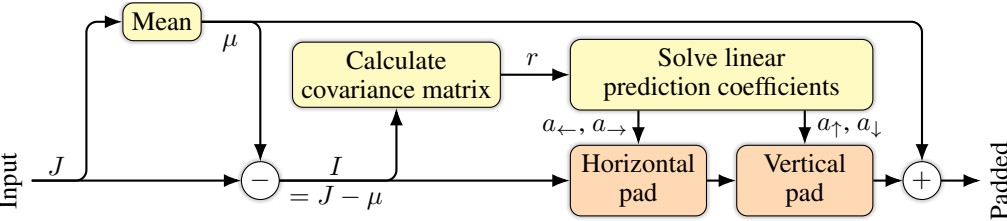

Figure 5: Linear prediction padding process for single-channel 2D data. The covariance and autocorrelation methods differ only in the *Calculate covariance matrix* block. Coefficients $a$ are solved for centered as well as for off-center prediction for corners, for each direction indicated.

## 2.1 Covariance method

For one ($P = 1$) and two-pixel ($P = 2$) neighborhoods, the error to minimize can be expressed using the shorthand $r_{ij}$ for the means of products that are elements of a *covariance matrix* $r$:

$$P{=}1: \quad \text{MSE} = \frac{1}{|S|} \sum_{(y,x) \in S} \left( I_{(y,x)+h_0} - a_1 I_{(y,x)+h_1} \right)^2 = a_1^2 r_{11} - 2a_1 r_{01} + r_{00}$$

$$P{=}2: \quad \text{MSE} = \frac{1}{|S|} \sum_{(y,x) \in S} \left( I_{(y,x)+h_0} - a_1 I_{(y,x)+h_1} - a_2 I_{(y,x)+h_2} \right)^2 \tag{5}$$

$$= a_1^2 r_{11} + 2a_1 a_2 r_{12} - 2a_1 r_{01} + a_2^2 r_{22} - 2a_2 r_{02} + r_{00},$$

$$\text{where } r_{ij} = \frac{1}{|S|} \sum_{(y,x) \in S} I_{(y,x)+h_i} I_{(y,x)+h_j}, \quad r_{ij} = r_{ji}. \tag{6}$$

The LS solutions can be found by solving what are known as the *normal equations*:

$$P{=}1: \quad \frac{\partial \text{MSE}}{\partial a_1} = 2a_1 r_{11} - 2r_{01} = 0 \quad \Rightarrow \quad a_1 = \frac{r_{01}}{r_{11}}$$

$$P{=}2: \quad \begin{cases} \frac{\partial \text{MSE}}{\partial a_1} = 2a_1 r_{11} + 2a_2 r_{12} - 2r_{01} = 0 \\ \frac{\partial \text{MSE}}{\partial a_2} = 2a_1 r_{12} + 2a_2 r_{22} - 2r_{02} = 0 \end{cases} \Rightarrow \begin{cases} a_1 = \frac{r_{01} r_{22} - r_{02} r_{12}}{r_{11} r_{22} - r_{12}^2} \\ a_2 = \frac{r_{02} r_{11} - r_{01} r_{12}}{r_{11} r_{22} - r_{12}^2}. \end{cases} \tag{7}$$

In implementation, we used a safe version of the division operator and its derivatives that replaces infinities with zeros in results. For any $P$, the normal equations involve the covariance matrix $r$:

$$\begin{bmatrix} r_{11} & r_{12} & \dots & r_{1P} \\ r_{12} & r_{22} & \dots & r_{2P} \\ \vdots & \vdots & \ddots & \vdots \\ r_{1P} & r_{2P} & \dots & r_{PP} \end{bmatrix} \begin{bmatrix} a_1 \\ a_2 \\ \vdots \\ a_P \end{bmatrix} = \begin{bmatrix} r_{01} \\ r_{02} \\ \vdots \\ r_{0P} \end{bmatrix}. \tag{8}$$

The approach is known as the *covariance method*. We implemented it only for the neighborhoods $1 \times 1$ (method `lp1x1cs` where `cs` stands for covariance, stabilized) and $2 \times 1$ (`lp2x1cs`), using Eq. 7 and stabilization of the effectively 1D linear predictors by reciprocating the magnitude of each below-unity-magnitude root of the AR process characteristic polynomial ($1 - a_1 B$ for the $1 \times 1$ neighborhood and $1 - a_1 B - a_2 B^2$ for $2 \times 2$) of lag operator $B$ and by obtaining the coefficients from the expanded manipulated polynomial (see Appendix B for details).

## 2.2 Autocorrelation method with Tukey window and zero padding

For methods `lp2x1`, `lp2x3`, `lp2x5`, `lp3x3`, `lp4x5`, and `lp6x7`, we redefined $r_{ij}$ as normalized $(N_y, N_x)$-periodic autocorrelation:

$$r_{ij} = \frac{R_{h_i - h_j}}{N_y N_x}, \quad R_{(\Delta y, \Delta x)} = \sum_{y=0}^{N_y - 1} \sum_{x=0}^{N_x - 1} I_{(y,x)} I_{\left( (y - \Delta y) \bmod N_y, (x - \Delta x) \bmod N_x \right)}. \tag{9}$$

To reduce periodization artifacts, for use in Eq. 9, we multiplied the zero-mean image horizontally and vertically by a Tukey window with a constant segment length of 50% and zero padded it sufficiently to prevent wraparound. For `lp2x5`, `lp3x3`, `lp4x5`, and `lp6x7` we accelerated calculation of $R$ using fast Fourier transforms (FFTs) and the Wiener–Khinchin theorem, $R = \text{IDFT2}\left( |\text{DFT2}(I)|^2 \right)$ for our purposes, where DFT2 and IDFT2 are the 2D discrete Fourier transform and its inverse.

## 2.3 Implementation

We implemented linear prediction padding in the JAX framework (Bradbury et al. 2025) and solved Eq. 8 using a differentiable Cholesky solver from Lineax (Rader et al. 2023), stabilizing solves by adding a small constant $10^{-7}$ to diagonal elements of the covariance matrix, i.e. by ridge regression.

While not dictated by the theory, we constrained our implementation to rectangular neighborhoods (as in Fig. 3) with the predicted pixel located adjacent to and centered on the neighborhood with the exception of corner padding (see Fig. 4). Our implementation benefited from the capability

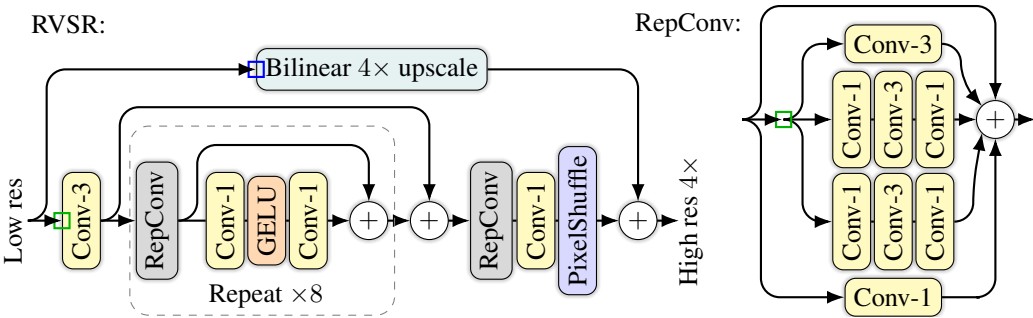

Figure 6: The convolutional RVSR super-resolution model. The spatial sizes of convolution kernels were $N \times N$ for Conv-$N$. Bilinear image upscaling methods in JAX and PyTorch implicitly replication pad their inputs. We padded upscale input explicitly (□) with the method configurable separately from padding of Conv inputs (□). The RepConv block was converted to a single Conv-3 for inference.

of JAX to 1) *scan* the recursion for paddings larger than a 1-pixel layer, 2) to *vmap* (vectorizing map) for parallelization of the padding front and for calculations over axes of rectangular unions of extended neighborhoods including corner handling variants, and 3) to fuse convolution with covariance statistics collection. As far as the authors are aware, such automated fusing would not take place in PyTorch that at present only offers an optimized computational kernel for zero padding.

## 3    Evaluation in RVSR super-resolution

We reimplemented the convolutional RVSR $4\times$ super-resolution model (Conde et al. 2024) in JAX-based Equinox (Kidger and Garcia 2021) with a fully configurable padding method (see Fig. 6)) and trained its 218 928 training-time parameters from scratch using MSE loss. For some experiments, to emulate network output center cropping, we omitted padding from the inputs of a number of the last conv layers and cropped the bilinear upscale. We define *output crop* as the number of 4-pixels-thick *shells* discarded in center cropping (see Fig.7). We used the same padding method in the convolutional and upscale paths, with the exception of zero-repl and zero-zero where the latter designator signifies the upscale padding method. The upscale output was cropped identically to output crop, rendering zero-repl and zero-zero equivalent (denoted zero) for output crop $\geq 1$.

We used a dataset of 10k $512\times512$ pixel Sentinel 2 Level-1C RGB images (Niemitalo et al. 2024. We linearly mapped reflectances from $[0, 1]$ to $[-1, 1]$. We split the data into a training set of 9k images and a test set of 1k images. The mean over images and channels of the variance of the test set was 0.44 with RGB means -0.28, -0.31, and -0.22. To assemble a training batch we randomly picked images without replacement from the set, randomly cropped each image to $200\times200$, created a low-resolution version by bilinearly downscaling with anti-aliasing to $50\times50$, and center cropped the images to remove edge effects resulting in $192\times192$ target images and $48\times48$ input images.

We used a batch size of 64 and the Adam optimizer (Kingma 2014) with $\varepsilon = 10^{-3}$ (increased to improve stability) and default $b_1 = 0.9$ and $b_2 = 0.999$, and the learning rate linearly ramped from $5 \times 10^{-6}$ to 0.014 over steps 0 to 100 (warmup, tuned for a low failure rate without sacrificing learning rate much) and from 0.014 to 0 over steps 1M to 1.5M (cooldown). We repeated the training with up to 12 different random number generator seeds.

For trained models, we also evaluated test MSE separately for shells #0–10, including in shell #10 also the rest of the shift-equivariantly processed output center. We report bootstrapped 95% confidence intervals over seeds for mean MSE and mean relative MSE difference. Test loss was calculated using center-cropped dataset images during training and by cropping at each corner in final evaluation.

Each training run took ∼4 days on a single NVIDIA V100 32G GPU, ∼2.5 GPU years in total consuming ∼5 MWh of 100% renewable energy. Development and testing consumed ∼15% additional compute. We used a V100S 32G to evaluate final MSEs, and an RTX 4070 Ti 16G for maximum batch size binary search and GPU throughput measurement at maximum batch size.

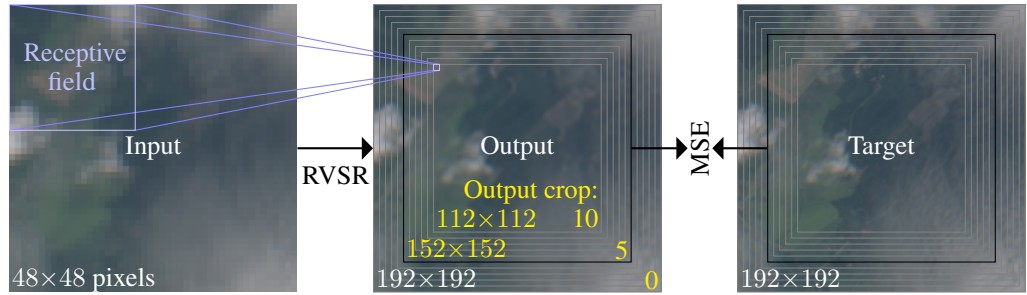

Figure 7: Illustration of RVSR super-resolution output cropping in MSE calculation, showing output crop 5 as an example. At output crop 10, the theoretical receptive fields do not include any padding.

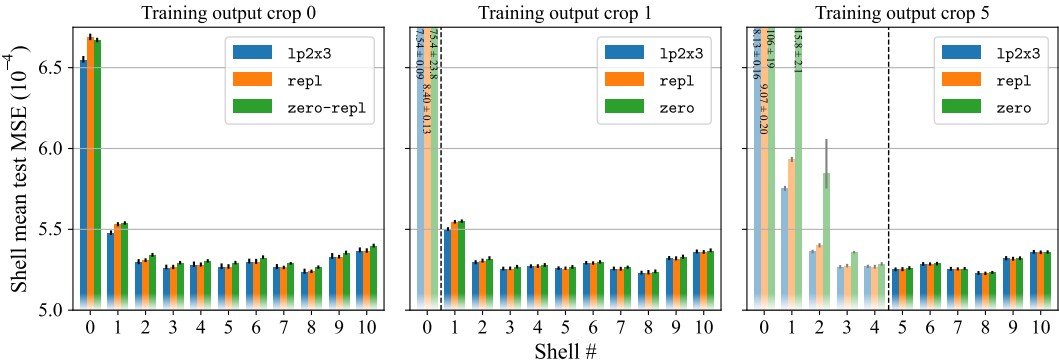

Figure 8: Super-resolution mean test MSE for select models, calculated separately for shells of equivalent width of 1 input pixel (bounded by squares in Fig. 7). Error bars indicate 95% confidence interval of the mean. Results for each output crop only include those seeds for which training was successful for every method listed. Pale-colored bars indicate exclusion of shells from training loss.

## 4    Results and discussion

For sample images padded using variants of `lp` and other methods, see Appendix D. Super-resolution evaluation results for trained models can be found in Table 1 and Figs. 8, 9, and 10. For sample super-resolution images and a visualization of the deviation from shift equivariance see Appendix G.

In RVSR super-resolution training, we observed catastrophic Adam optimizer instability with some seed–method combinations (see Appendix F). We believe that this was a chaotic effect due to both a high learning rate and to numerical issues exemplified by differences between the equivalent `repl` and `extr1`, and not indicative of an inherent difference in training stability between the various padding methods. Overall training dynamics (Appendix C) were similar between the padding methods, with the exception of lower early-training test MSE for `repl` and the `lp` methods, compared to a `zero-repl` baseline, when trained without output crop.

The different `lp` methods yielded very similar super-resolution test MSEs, bringing the light-weight `lp1x1cs` and `lp2x1cs` to the inference throughput–MSE Pareto front (see Fig. 10) for each output crop 0 and 1. The `lp` methods that used FFT-accelerated autocorrelation reached larger batch sizes in training but were limited to smaller batch sizes in inference, in comparison to `lp` methods that calculated autocorrelation directly and used a similar neighborhood size.

For output crop 0, every tested `lp` method yielded a 0.20–0.9% (at 95% confidence) lower mean test MSE and 1.5–2.0% lower outermost shell mean test MSE compared to the standard `zero-repl` baseline (see Tab. 1). For the other shells (see Fig. 8), `lp2x3` and `repl` were tied but consistently better than the baseline. Compared to the baseline, `repl` improved the test mean MSE by 0.1–0.7% but, notably, was 0.1–0.5% worse at the outermost shell. We hypothesize that the more clear edge signal and worse data approximation by `zero-repl`, compared to `repl`, enables and forces the network to use some of its capacity to improve shell #0 performance at the cost of overall performance. As an approximator of the internal feature maps of a trained RVSR network (with GELU activation),

| Output crop | Conv and upscale padding method(s) | FFT ($\mathcal{F}$) or direct (D) autocorrelation | Maximum training batch size (images) ↑ | Training throughput (images/s) ↑ | Maximum inference batch size (images) ↑ | Inference throughput ($10^6$ pixels/s) ↑ | Mean test MSE ($10^{-6}$) ↓ | Mean test MSE diff to zero-repl (%) → | Outermost shell mean test MSE diff to zero-repl (%) → |
|---|---|---|---|---|---|---|---|---|---|
| 0 | extr1 | | **266** | 480 | 6741 | 865 | 545.0 ± 2.4 | −0.25 ± 0.31 | 0.43 ± 0.24 |
| | extr2 | | **266** | 480 | 6741 | 759 | 550.5 ± 2.9 | 0.77 ± 0.35 | 5.67 ± 0.53 |
| | extr3 | | **266** | 478 | 6741 | 758 | 559.6 ± 1.9 | 2.38 ± 0.39 | 11.98 ± 0.40 |
| | lp1x1cs | | 237 | 464 | 6741 | 722 | 543.3 ± 3.3 | −0.64 ± 0.39 | −1.82 ± 0.28 |
| | lp2x1 | D | 190 | 433 | 6741 | 498 | 543.2 ± 2.3 | −0.63 ± 0.25 | −1.71 ± 0.19 |
| | lp2x1cs | | 189 | 442 | 6741 | 632 | 542.9 ± 2.9 | −0.68 ± 0.28 | −1.88 ± 0.14 |
| | lp2x3 | D | 187 | 429 | 6741 | 480 | 542.8 ± 3.0 | −0.71 ± 0.19 | −1.90 ± 0.18 |
| | lp2x5 | $\mathcal{F}$ | 240 | 453 | 6020 | 382 | **542.7 ± 2.5** | **−0.72 ± 0.17** | **−1.90 ± 0.11** |
| | lp3x3 | $\mathcal{F}$ | 241 | 455 | 6020 | 394 | 543.2 ± 2.5 | −0.65 ± 0.25 | −1.89 ± 0.12 |
| | lp4x5 | $\mathcal{F}$ | 236 | 447 | 6020 | 356 | 543.2 ± 3.5 | −0.63 ± 0.39 | −1.77 ± 0.18 |
| | lp6x7 | $\mathcal{F}$ | 191 | 432 | 6020 | 266 | 543.7 ± 1.8 | −0.49 ± 0.23 | −1.34 ± 0.23 |
| | repl | | **266** | **481** | **6880** | 870 | 544.6 ± 3.1 | −0.39 ± 0.30 | 0.28 ± 0.17 |
| | zero-repl | | 263 | 472 | **6880** | **964** | 546.6 ± 1.9 | 0.00 | 0.00 |
| | zero-zero | | 263 | 472 | 6741 | 960 | 547.1 ± 1.8 | 0.09 ± 0.26 | 1.32 ± 0.15 |
| 1 | lp1x1cs | | 238 | 468 | **7314** | 695 | 532.2 ± 1.6 | **−0.27 ± 0.29** | **−0.93 ± 0.16** |
| | lp2x1cs | | 191 | 444 | **7314** | 608 | **532.0 ± 1.8** | −0.27 ± 0.17 | −0.88 ± 0.20 |
| | lp2x3 | D | 188 | 436 | **7314** | 473 | 532.5 ± 2.3 | −0.22 ± 0.26 | −0.88 ± 0.13 |
| | repl | | **268** | **481** | **7314** | 808 | 532.8 ± 2.6 | −0.16 ± 0.31 | −0.07 ± 0.13 |
| | zero | | **268** | **483** | **7314** | **896** | 533.6 ± 2.0 | 0.00 | 0.00 |
| 5 | lp2x3 | D | 257 | 486 | **8403** | 465 | 531.8 ± 2.8 | −0.05 ± 0.30 | **−0.16 ± 0.14** |
| | repl | | **325** | 518 | **8403** | 664 | **531.7 ± 2.5** | **−0.06 ± 0.27** | −0.14 ± 0.17 |
| | zero | | **325** | 518 | **8403** | **706** | 532.0 ± 2.2 | 0.00 | 0.00 |

Table 1: RVSR evaluation results. 95% convidence intervals are reported (gray means an insufficient number of runs). The best results for each output crop are in **bold**. For output crop 0 we tested zero padding of convolution inputs (method names beginning with `zero-`) together with both zero and replication padding of upscale inputs (indicated after the hyphen). Overall, linear prediction methods had the lowest test MSE while `repl` and `zero` were the fastest and used the least memory.

`zero` has up to 40-fold larger error than `repl` and `lp2x3` (see Fig. 11). The worse approximation by `zero` is illustrated by the up to 10-fold MSE error growth for shells not included in the training loss and outside the output crop (shaded gray in Fig. 8), compared to `repl` and `lp2x3`. Compared to explicit `zero`, the default implicit `repl` in bilinear upscale gave a lower outermost shell mean test MSE.

For output crop 1, the choice of padding method mattered less, with `lp2x3` improving upon the baseline at the two outermost shells. Models trained with output crop 5 saw no difference from the choice of padding method. Furthermore, the test MSEs have only relatively modest differences between output crops. At shells #$\geq$5, all models perform similarly with the exception of the somewhat worse output crop 0 `zero-repl`. This might be because training with output crop doesn't free up sufficient capacity to decrease MSE in the remaining image area for `repl` and `lp2x3`.

For $\text{extr}N$ we found test MSE to increase with $N$, with `extr3` giving 11–12% worse outermost shell mean MSE compared to baseline. In contrast, Leng and Thiyagalingam 2023a found the equivalent of `extr3` (see GitHub issue #2 in Leng and Thiyagalingam 2023b for a numerical demonstration of equivalence) to give better results in a U-net super-resolution task than `zero` or `repl`, which we suspect was due to their use of blurred inputs (Gaussian blur of standard deviation $\sigma = 3$) that are better approximated by the higher-degree `extr3` method (see Appendix E).

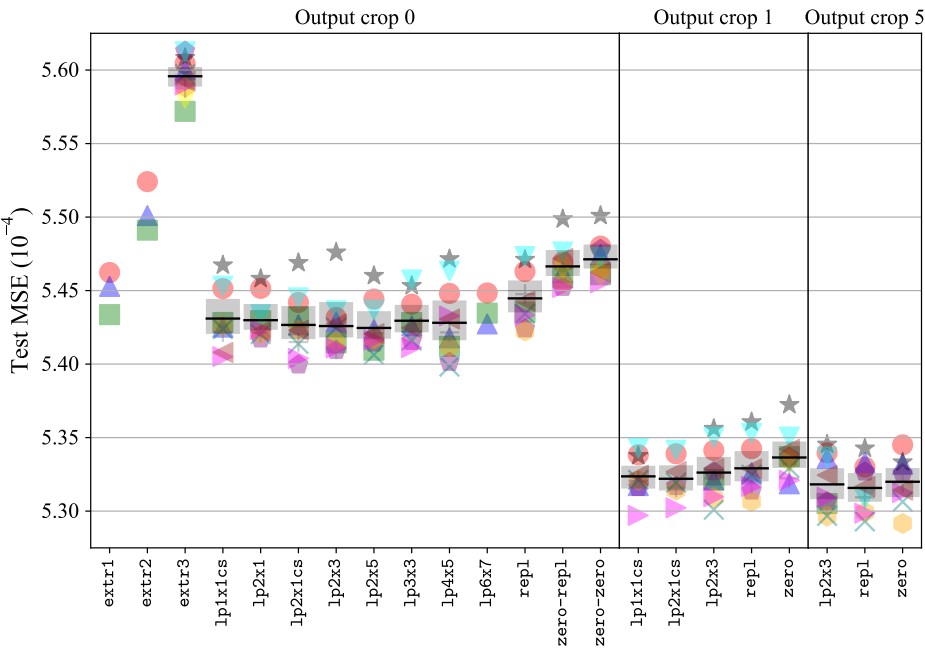

Figure 9: Super-resolution test MSE with mean (—) and the 95% confidence interval of the mean (▬) across all successful training runs. For a tabular presentation and seed values, see Appendix F.

## 5    Conclusions

Using linear prediction padding (`lp`) instead of `zero` or replication padding (`repl`) improved slightly the quality of CNN-based super-resolution, in particular near image borders, at a moderate added time cost. Center-cropping the network output leveled the differences in output–target mean square error between padding methods. At output crop 5 the stitching artifacts due to deviation from shift equivariance were no longer visible.

Considering padding as autoregressive estimation of data and feature maps explains some of the differences between padding methods. However, the tested CNN architecture learned to compensate the elevated super-resolution error near the image edge to roughly the same magnitude for all the tested padding methods, including `zero` which has an exceptionally high estimation error. The slightly higher overall super-resolution error with `zero` supports the hypothesis that more network capacity is consumed by the compensation of the larger padding error.

Our results might not directly apply to other CNN architectures and tasks. Covariance statistics may suffer from the small sample problem with spatially tiny inputs such as encodings. Larger effective receptive fields may favor `lp`, whereas workloads with a higher level of spatial inhomogeneity, lower spatial correlations in network input or feature maps (in particular spatially whitened data), or higher nonlinearity in spatial dependencies would likely make `lp` less useful, favoring `zero` for its clear edge signaling. If using `lp` in CNN-based processing of images with *framing*, for example photos of objects, any needed location information might need to come from another source than the padding.

Our JAX `lp` padding implementation and source code for reproducing the results of this article are included in the enclosed zip file. Our `lp1x1cs` and `lp2x1cs` methods would be the most straightforward ones to port to other frameworks.

## 6    Further work

We have yet to explore 1) using a spatially weighted loss to level spatial differences in error, 2) using batch rather than image statistics for a larger statistical sample, 3) increasing the sample size by giving `lp` memory of past statistics, 4) learning rather than solving `lp` coefficients, 5) modeling dependencies between channels, 6) instead of ouput cropping, cross-fading adjacent output tiles and

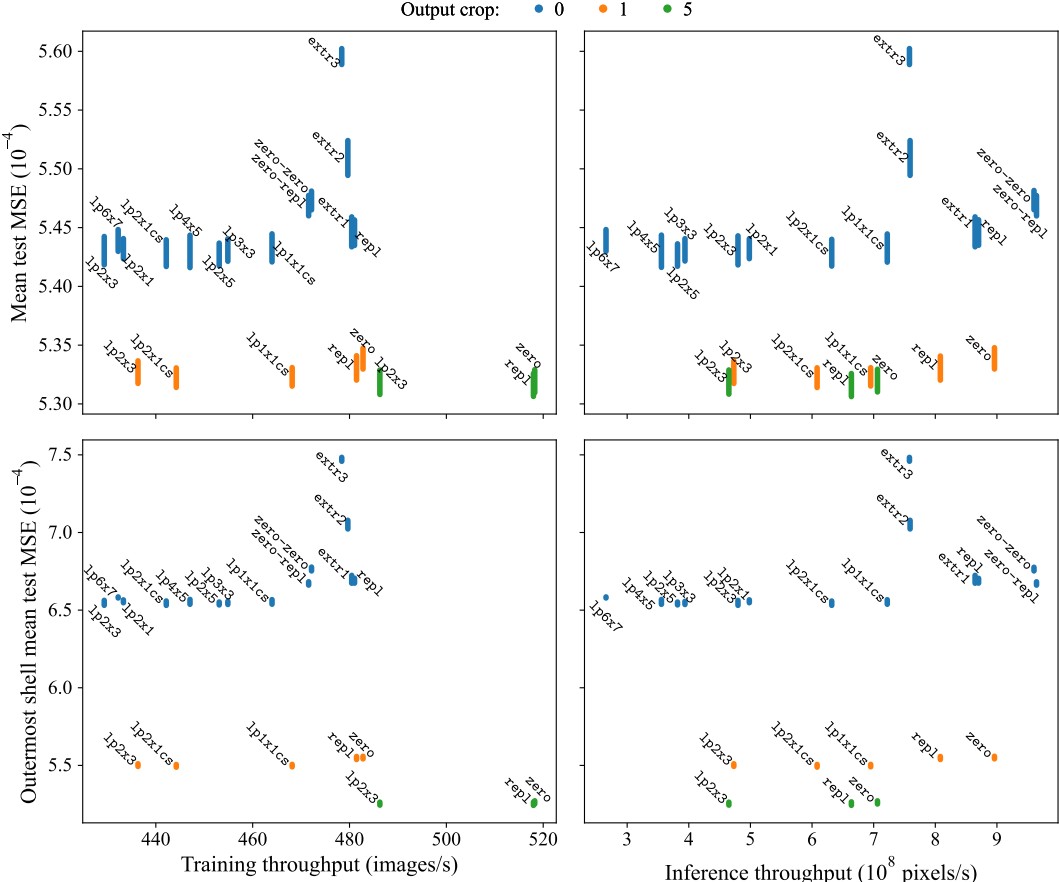

Figure 10: 95% confidence interval of the mean super-resolution MSE of each model vs. training/inference throughput evaluated using maximum training/inference batch sizes. Except for the outermost shell mean test MSE, the results do not apply to other than the used $48{\times}48$ pixel input image size because of the difference in the ratio of output pixels influenced by padding.

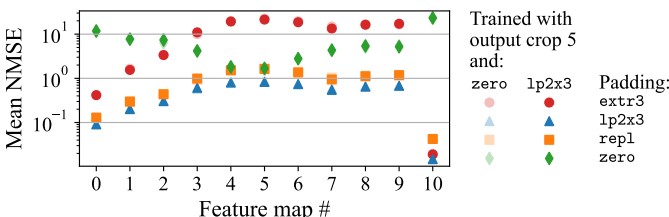

Figure 11: Test mean-over-seeds NMSE (MSE divided by data variance) of padding each convolution layer input (low-res input #0 and feature maps #1–9) and high-res network output (#10) from a $28{\times}28$-pixel center crop to $30{\times}30$ pixels, in a trained RVSR network. Padding by mean would give NMSE $= 1$. Sparsity differences (Aimar et al. 2018) may contribute to predictability differences.

optimizing the cross-fade curves during training, 7) setting the padding method separately for each feature map based on its spatial autocorrelation, 8) accelerating solves by taking advantage of the covariance matrix structure, and 9) use of `lp` in other CNN architectures and training settings.

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

# A  A triangle-inequality-like inequality for squared distances

This section gives a derivation of the result that the mean square deviation between two vectors of equal length is bounded from above by four times their average mean square deviation from a third vector of equal length.

Let $A$, $B$, and $C$ be length-$N$ vectors of real numbers. Denoting the sum of squares of elements of vector $x$ by $||x||^2$, the parallelogram law (see Jordan and Neumann [1935]) for $x = A - C$ and $y = C - B$ is:

$$||x + y||^2 + ||x - y||^2 = 2\left(||x||^2 + ||y||^2\right)$$
$$\iff \quad ||A - C + C - B||^2 + ||A - C - C + B||^2 = 2\left(||A - C||^2 + ||C - B||^2\right) \qquad (10)$$
$$\iff \quad ||A - B||^2 + ||A - 2C + B||^2 = 2\left(||A - C||^2 + ||C - B||^2\right).$$

By subtracting the non-negative $||A - 2C + B||^2$ from the left side we get an inequality similar to the triangle inequality, but for squared distances rather than distances:

$$||A - B||^2 \leq 2\left(||A - C||^2 + ||B - C||^2\right). \qquad (11)$$

By multiplying both sides by $\frac{1}{N}$ with $N$ the common length of the vectors:

$$\frac{1}{N}||A - B||^2 \leq 2\left(\frac{1}{N}||A - C||^2 + \frac{1}{N}||B - C||^2\right), \qquad (12)$$

and by identifying $\frac{1}{N}||x - y||^2$ as the mean square deviation (MSD) between $x$ and $y$, we can write:

$$\text{MSD}(A, B) \leq 2\left(\text{MSD}(A, C) + \text{MSD}(B, C)\right)$$
$$\iff \quad \text{MSD}(A, B) \leq 4\,\text{mean}\big(\text{MSD}(A, C),\, \text{MSD}(B, C)\big). \qquad (13)$$

This means that the mean square deviation between two predictions $A$ and $B$ is bounded from above by four times their average mean square prediction error with $C$ as the ground truth.

# B  Stabilization of 1D covariance method linear prediction

For 1D linear prediction neighborhoods 1×1 (stabilized covariance method `lp1x1cs`) and 2×1 `lp2x1cs`), the padding procedure in one direction (Eq. 4) using already calculated coefficients $a_{1...P}$ is equivalent to a discrete-time linear time-invariant (LTI) system having a causal recursion:

$$1{\times}1:\ \ y[k] = a_1 y[k-1] + b_0 x[k], \qquad 2{\times}1:\ \ y[k] = a_1 y[k-1] + a_2 y[k-2] + b_0 x[k], \quad (14)$$

where $y[k]$ are known pixel values or padding pixels, $x[k]$ are input pixels with $x[k] = 0$ for all $k$ with an inconsequential input coefficient $b_0$. The corresponding transfer functions are:

$$
\begin{aligned}
1{\times}1:\quad H(z) &= \frac{b_0}{1 - a_1\, z^{-1}} & 2{\times}1:\quad H(z) &= \frac{b_0}{1 - a_1\, z^{-1} - a_2\, z^{-2}} \\
&= \frac{b_0\, z}{z - p_0}, & &= \frac{b_0\, z^2}{(z - p_0)(z - p_1)}, \\
p_0 &= a_1 & p_0 &= \frac{a_1 + \sqrt{a_1^2 + 4\, a_2}}{2} \\
& & p_1 &= \frac{a_1 - \sqrt{a_1^2 + 4\, a_2}}{2},
\end{aligned}
\qquad (15)
$$

where $b_0$ is an input scaling factor, $z^{-1}$ represents a delay of one sampling period, and $H(e^{i\omega})$, represents the frequency response of the system with $\omega$ the frequency in radians per sampling period, $e$ the natural number, and $i$ the imaginary unit.

The system is stable if all poles $p$ of the transfer function lie inside the complex $z$-plane unit circle. A stationary autoregressive process is stable. If the coefficients were found by solving normal equations with approximate covariances, then stability is not guaranteed. In practice, stability is needed to prevent blow-up of the padding output when padding recursively.

By reciprocating the $z$-plane radius of all poles that have radius $> 1$ (i.e. $|p| > 1$), the system can be made stable, or marginally stable in case any of the poles lie at radius 1 exactly. An unstable system has no well-defined frequency response, but we can still compute $H(e^{i\omega})$. The stabilization alters the phase of $H(e^{i\omega})$ but maintains its magnitude up to a constant scaling factor that is inconsequential with zero input, thus preserving the essential power-spectral characteristics of the autoregressive process. Magnitude scaling could be compensated for by setting $b'_0 = b_0\big(1 - \sum_{i=0}^{i<P} a'_i\big)/\big(1 - \sum_{i=0}^{i<P} a_i\big)$ where $'$ denotes updated variables. The choice of $b_0$ is inconsequential with constant zero input but would matter in *generative* padding with $x$ a white noise *innovation*.

For a $2 \times 1$ neighborhood, if what's under the square root in Eq. 15 is negative, $a_1^2 + 4a_2 < 0$, then the poles are complex and form a complex conjugate pair. Otherwise, both poles are real. The squared magnitudes of the complex conjugate pair of poles are equal, $|p_0|^2 = |p_1|^2 = -a_2$. The complex poles lie outside the unit circle only if $-a_2 > 1$, in which case the system can be stabilized by $a'_1 = -a_1/a_2$, $a'_2 = 1/a_1$. Real poles $p_0$ and $p_1$ can be found using Eq. 15, they can be reciprocated when necessary, and the modified coefficients can be extracted from the expanded form of the numerator polynomial as: $a'_1 = p'_0 + p'_1$ and $a'_2 = p'_0 \, p'_1$.

The characteristic polynomial of the AR process is the denominator of the transfer function Eq. 15 written with lag operator $B = z^{-1}$. With the characteristic polynomial the stability condition is that all roots are outside the unit circle. Stabilization via the characteristic polynomial would manipulate the coefficients identically to what was presented above.

## C   RVSR super-resolution training loss histories

Some RVSR super-resolution training loss histories averaged over seeds are shown in Fig. 12

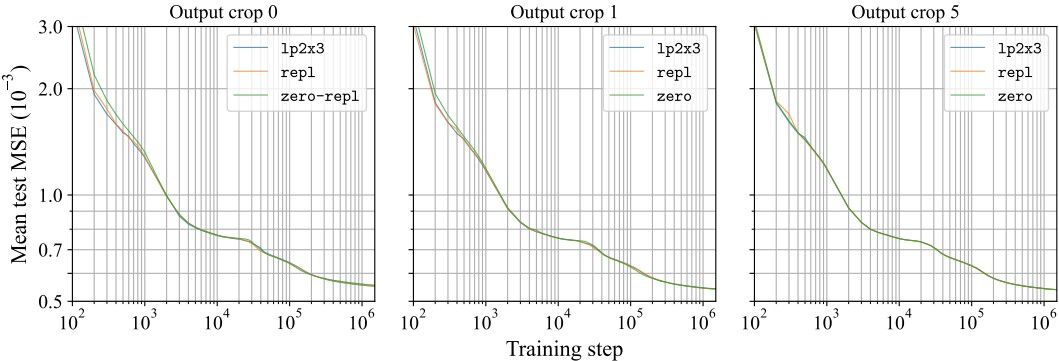

Figure 12: Super-resolution mean test loss during training for select padding methods, including for each output crop only those seeds that resulted in a successful training run for every method listed.

## D  Sample padded images

Fig. 13 shows sample padded images. For information about the image dataset, see section 3.

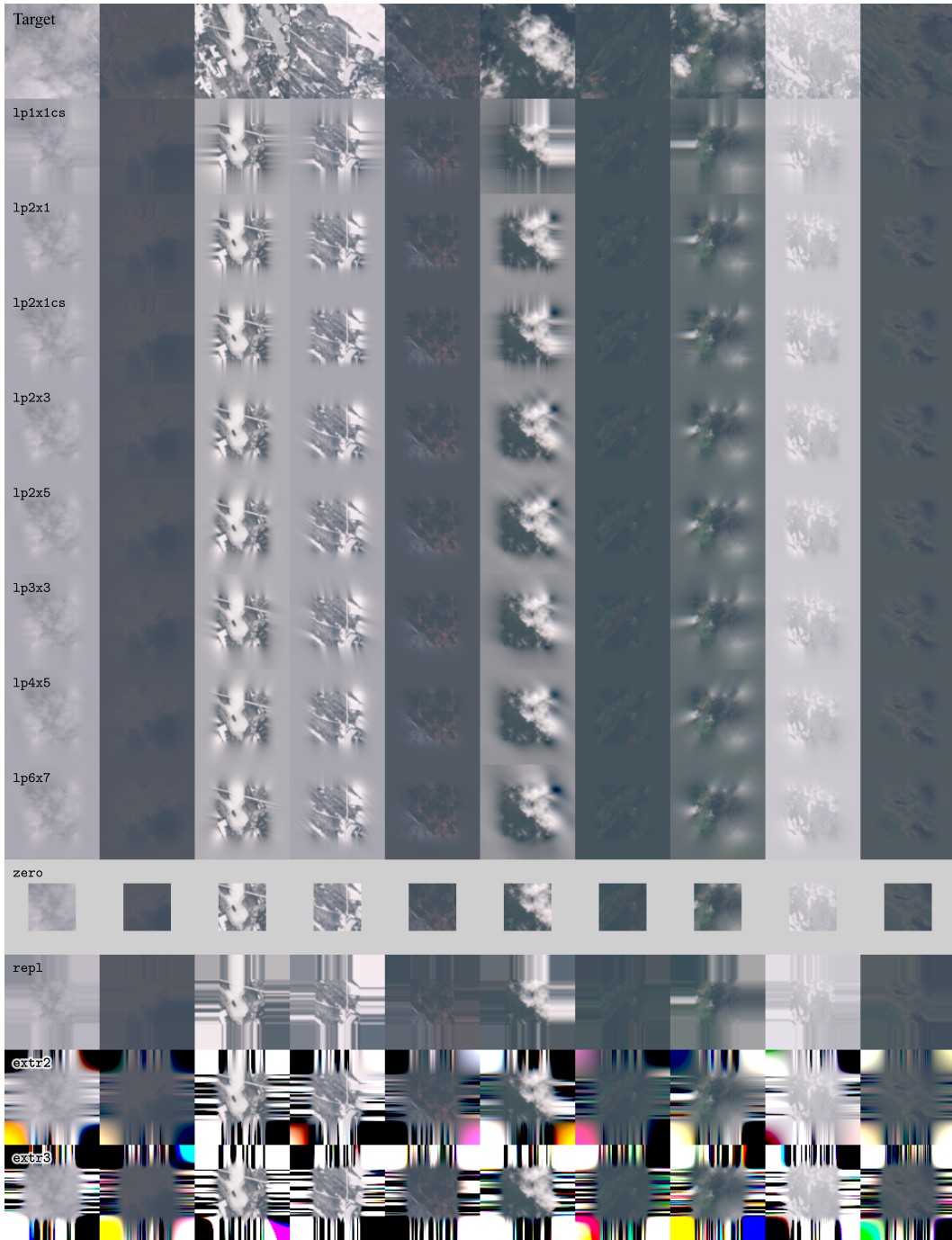

Figure 13: Non-cherry-picked $48 \times 48$ pixel test set satellite images padded with $24$ pixels on each side, using different methods. The first row shows the ground truth (*target*) image from which each padding input was center-cropped (see the `zero` row for the crop boundaries). The `lp` methods with larger neighborhood widths capture directional regularities with larger slopes off the padding direction. The `extr2` and `extr3` recursions are unstable. Color channels have been clipped to reflectance range 0–0.8.

# E   Effect of blur on padding error

To simulate padding input data having an adjustable degree of blurriness or a rate of frequency
spectral decay, we model Gaussian-blurred white-noise data by uniformly sampling a zero-mean
Gaussian process $x$ of unit variance and with Gaussian covariance as function of lag $d$:

$$\text{Cov}(x[i], x[i+d]) = \kappa(d) = \exp\left(-\frac{d^2}{2\sigma^2}\right) \quad \text{for all integer } i, \tag{16}$$

with $\sigma$ corresponding to the standard deviation of the Gaussian blur. A general linear right padding
method approximates $x[0]$ from the $P$ nearest samples by $\hat{x}[0] = \sum_{i=1}^{P} a_i x[-i]$. We define the
padding error $\varepsilon$ by:

$$\varepsilon = x[0] - \hat{x}[0] = -\sum_{i=0}^{P} a_i x[-i], \quad a_0 = -1, \tag{17}$$

prepending $a_{1\ldots P}$ with $a_0 = -1$ for convenience. The normalized mean square of the zero-mean $\varepsilon$ is:

$$\text{NMSE} = \frac{\text{E}(\varepsilon^2)}{\text{Var}(x)} = \frac{\text{Var}(\varepsilon)}{1} = \sum_{i=0}^{P}\sum_{j=0}^{P} \text{Cov}(a_i x[-i], a_j x[-j]) = \sum_{i=0}^{P}\sum_{j=0}^{P} a_i a_j \kappa(j-i) \tag{18}$$

We compare in Fig. 14 the MSE for the following 1D padding methods with some equivalencies:

| Method(s) | $P$ | $a_{0\ldots P}$ |
|---|---|---|
| `zero`, `extr0` | 0 | $-1$ |
| `repl`, `extr1` | 1 | $-1, 1$ |
| `extr2` | 2 | $-1, 2, -1$ |
| `extr3` | 3 | $-1, 3, -3, 1$ |
| `lp1x1cs` | 1 | $-1, a_1$ |
| `lp2x1`, `lp2x1cs` | 2 | $-1, a_1, a_2$ |
| `lp3x1` | 3 | $-1, a_1, a_2, a_3$ |

(19)

with $a_{1\ldots P}$ for `lp1x1cs`, `lp2x1`, and `lp2x1cs` from Eq. 7 and for `lp3x1` from solving Eq. 8 using
Levinson recursion, with known autocorrelation-like covariances $r_{ij} = \kappa(j-i)$, corresponding to the
limiting case of infinitely vast padding input providing covariance statistics matching the covariances
of the Gaussian process.

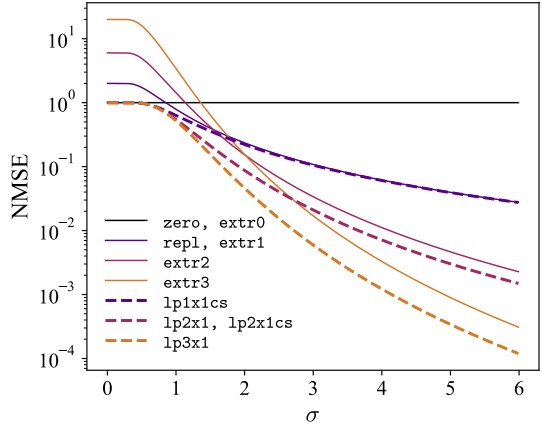

Figure 14: Theoretical padding NMSE as function of standard deviation $\sigma$ of Gaussian blur applied to
white-noise data. Padding input is the blurred data normalized to unit variance. Increasing the order
of the `extr` padding mode increases the error for low $\sigma$ and decreases it for high $\sigma$. Each `lp` method
has been least-squares fit to the known signal model and thus attains the lowest possible MSE given
the number of predictors. `zero` remains ideal for data with zero autocorrelation at non-zero lags.

# F  Trained models

| output crop | padding method | seed / symbol | | | | | | | | | | | |
|---|---|---|---|---|---|---|---|---|---|---|---|---|---|
| | | 0 ● | 1 ▲ | 2 ■ | 3 ◆ | 4 ⬟ | 5 ● | 6 ★ | 7 ✕ | 8 + | 9 ▼ | 10 ▶ | 11 ◀ |
| 0 | lp1x1cs | ✓ | ✓ | ✓ | ✗ | ✗ | ✓ | ✓ | ✓ | ✓ | ✓ | ✓ | ✓ |
| | lp2x1 | ✓ | ✗ | ✓ | ✗ | ✓ | ✓ | ✓ | ✓ | ✓ | ✓ | ✓ | ✓ |
| | lp2x1cs | ✓ | ✓ | ✓ | ✗ | ✓ | ✓ | ✓ | ✓ | ✓ | ✓ | ✓ | ✓ |
| | lp2x3 | ✓ | ✓ | ✓ | ✗ | ✓ | ✓ | ✓ | ✓ | ✓ | ✓ | ✓ | ✓ |
| | lp2x5 | ✓ | ✓ | ✓ | ✗ | ✓ | ✓ | ✓ | ✓ | ✓ | ✓ | ✓ | ✓ |
| | lp3x3 | ✓ | ✓ | ✓ | ✗ | ✓ | ✗ | ✓ | ✓ | ✓ | ✓ | ✓ | ✓ |
| | lp4x5 | ✓ | ✓ | ✓ | ✗ | ✓ | ✓ | ✓ | ✓ | ✓ | ✓ | ✓ | ✓ |
| | lp6x7 | ✓ | ✓ | ✓ | — | — | — | — | — | — | — | — | — |
| | zero-repl | ✓ | ✓ | ✓ | ✓ | ✓ | ✓ | ✓ | ✗ | ✗ | ✓ | ✓ | ✓ |
| | zero-zero | ✓ | ✓ | ✓ | ✓ | ✓ | ✓ | ✓ | ✗ | ✗ | ✓ | ✓ | ✓ |
| | repl | ✓ | ✗ | ✓ | ✗ | ✓ | ✓ | ✓ | ✓ | ✓ | ✓ | ✓ | ✓ |
| | extr1 | ✓ | ✓ | ✓ | — | — | — | — | — | — | — | — | — |
| | extr2 | ✓ | ✓ | ✓ | — | — | — | — | — | — | — | — | — |
| | extr3 | ✓ | ✓ | ✓ | ✓ | ✓ | ✓ | ✓ | ✓ | ✓ | ✓ | ✓ | ✓ |
| 1 | lp1x1cs | ✓ | ✓ | ✗ | ✗ | ✓ | ✓ | ✓ | ✓ | ✓ | ✓ | ✓ | ✓ |
| | lp2x1cs | ✓ | ✗ | ✗ | ✗ | ✓ | ✓ | ✗ | ✓ | ✓ | ✓ | ✓ | ✓ |
| | lp2x3 | ✓ | ✓ | ✓ | ✗ | ✓ | ✓ | ✓ | ✓ | ✓ | ✓ | ✓ | ✓ |
| | zero | ✗ | ✓ | ✓ | ✓ | ✓ | ✓ | ✓ | ✓ | ✓ | ✓ | ✓ | ✓ |
| | repl | ✓ | ✓ | ✗ | ✗ | ✓ | ✓ | ✓ | ✓ | ✓ | ✓ | ✓ | ✓ |
| 5 | lp2x3 | ✓ | ✓ | ✓ | ✗ | ✓ | ✓ | ✓ | ✓ | ✓ | ✓ | ✓ | ✓ |
| | zero | ✓ | ✓ | ✗ | ✗ | ✓ | ✓ | ✓ | ✓ | ✓ | ✗ | ✓ | ✓ |
| | repl | ✓ | ✓ | ✗ | ✗ | ✓ | ✓ | ✓ | ✓ | ✓ | ✓ | ✓ | ✓ |

Super-resolution training was either successful (✓), failed (✗), or skipped (−) for different combinations of padding methods and the random number generator seed. We report loss differences between models using only seeds that resulted in successful training for both of the compared models.

# G  Tiled processing samples and deviation from shift equivariance

Figs. 16–18 show non-cherry-picked stitched tiled processing results from models trained with seed 10, using different output crops. Fig. 15 show the corresponding inputs and targets from the test set. Figs. 19–21 visualize deviation from shift equivariance for the same images.

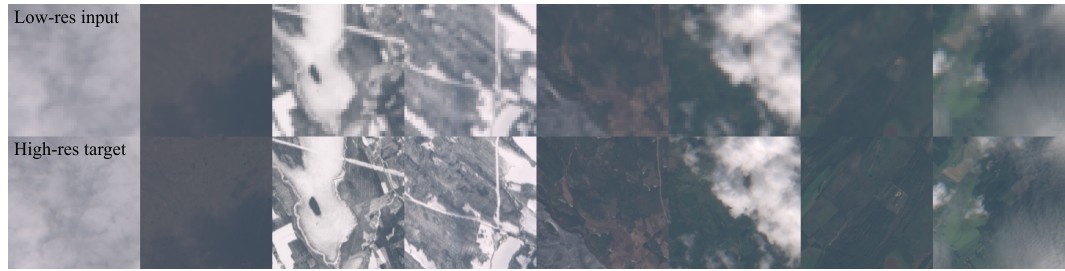

Figure 15: The super-resolution inputs and targets from the test set, cropped to the same view as the sample results.

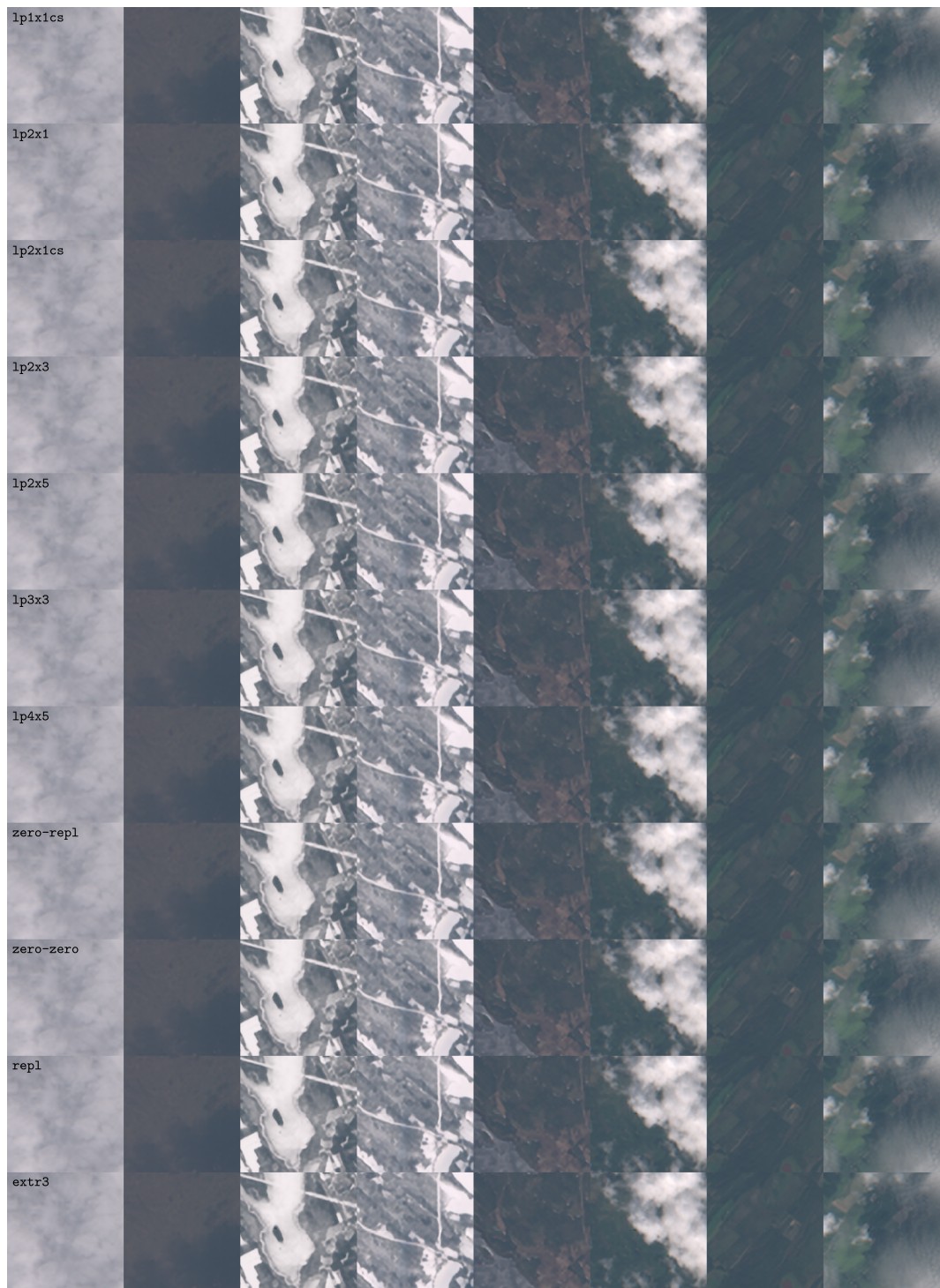

Figure 16: Stitched RVSR super-resolution results with output crop 0. Four output tiles, each as large as the images shown, were stitched together and cropped, with a corner of the tiling grid at the center of each image. A cross-hair-shaped discontinuity artifact is formed due to deviation from shift equivariance. Black represents reflectance 0 and white represents reflectance 0.8 on every channel.

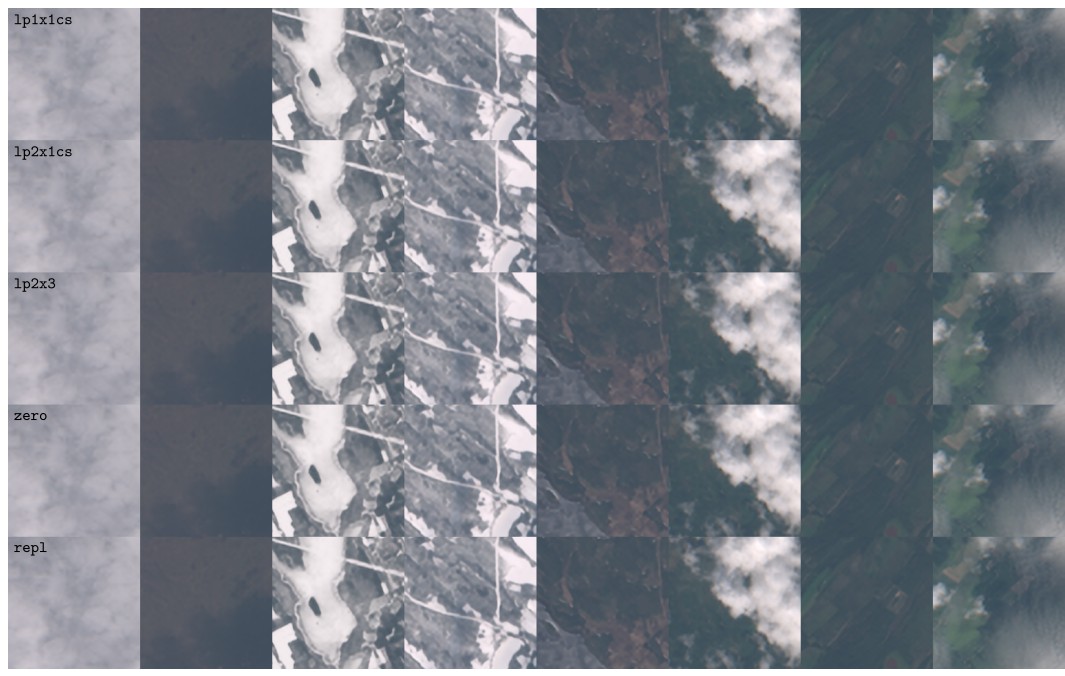

Figure 17: Stitched RVSR super-resolution results with output crop 1. Visually, the discontinuity artifact is much weaker than with output crop 1 and is only visible on high-contrast edges not perpendicular to the tile boundary.

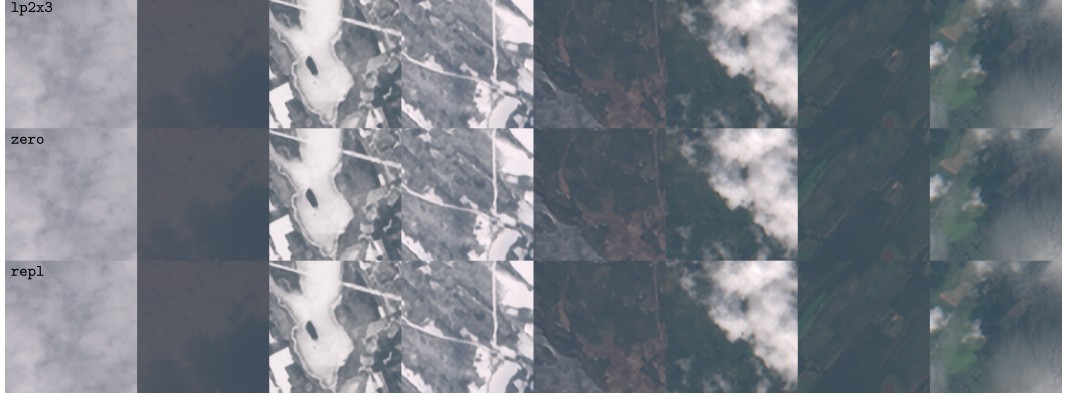

Figure 18: Stitched RVSR super-resolution results with output crop 5. No visible tiling artifacts.

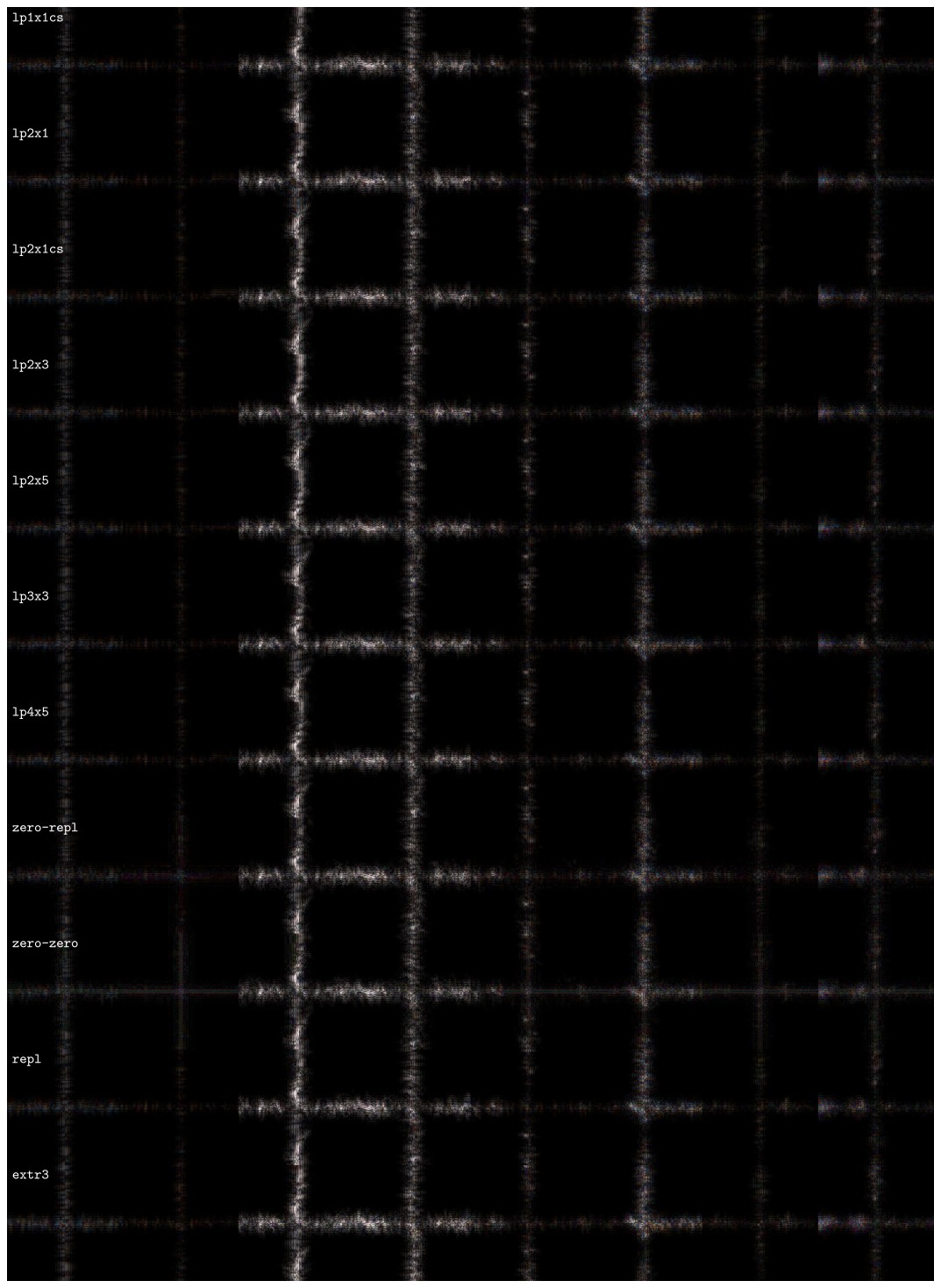

Figure 19: RVSR super-resolution deviation from shift equivariance for output crop 0, calculated as the absolute value of the difference between a stitched tiled prediction and a prediction using only valid convolutions. In these figures, black represents no difference and white represents an absolute reflectance difference of 0.2 on every channel.

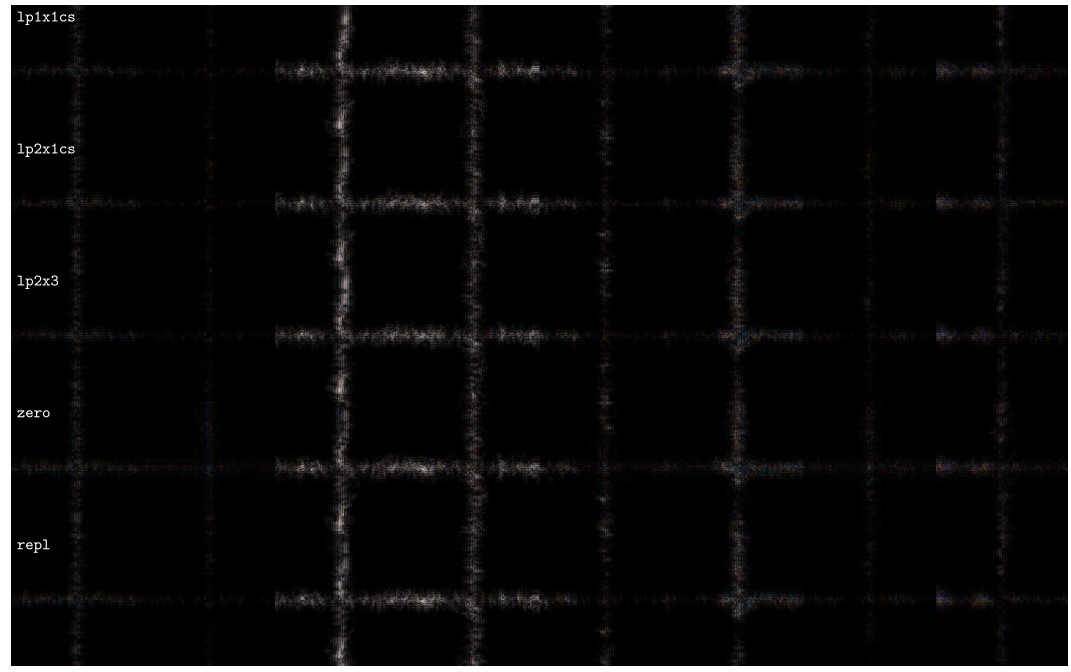

Figure 20: RVSR super-resolution deviation from shift equivariance for output crop 1.

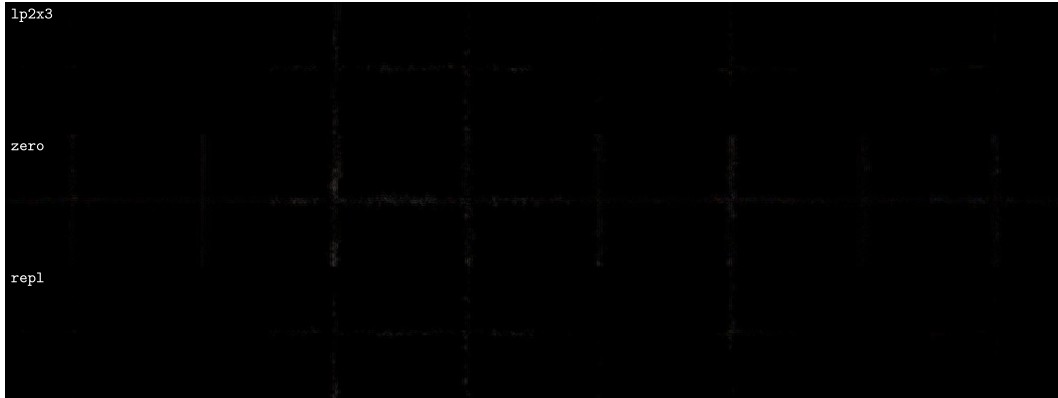

Figure 21: RVSR super-resolution deviation from shift equivariance for output crop 5. `zero` displays elevated deviations compared to `lp2x3` and `repl`.

