# OpenReview forum: "Per-channel autoregressive linear prediction padding in tiled CNN processing of 2D spatial data"
_NeurIPS.cc/2025/Conference — Submitted to NeurIPS 2025_

### Official Review · Reviewer_2k7F · 2025-06-17

**Clarity:** 1
**Significance:** 1
**Originality:** 2
**Rating:** 2
**Confidence:** 5

**Summary:**

The provided manuscript introduces a novel padding approach that utilizes linear predictions from an autoregressive linear model. The method's evaluation focuses on super-resolution, specifically tested on a single dataset containing 10k images of size 512x512. The super-resolution task is performed on an anti-aliased, downsampled version of this dataset. Results indicate a small improvement with the proposed padding.

**Questions:**

1. Why are the predicted pixels in Figure 4 (top-left) not actually at the top, but rather in the middle of the feature map/image? It appears the proposed method predicts padding values that should typically be at the corners/edges of the feature map/image.
2. In line 83, the authors state, "The approach is known as the covariance method." Could the authors please provide a reference for this claim?
3. Why are some results in Table1 marked in gray?
4. What do the different datapoints/shapes/colors represent in Figure 9? Please indicate by adding a legend or even improve visualization by using a different chart style like boxplots.

**Ethical Concerns:**

["NO or VERY MINOR ethics concerns only"]

**Final Justification:**

Thank you for the detailed rebuttal and for providing the additional analysis. While I appreciate the effort to clarify your position, my original concerns remain.

I understand the argument that the segmentation experiments were meant to showcase the generalizability of the proposed approach. However, the results show that the proposed approach fails to achieve competitive results compared to much simpler approaches like zero padding. The performance gap and the line of argumentation made by the authors on which segmentation tasks can be considered suggests that the core approach is not robust enough to generalize effectively, which is precisely my concern.

Further, the authors did not comment on the computational overhead of their proposed method, which is another major concern regarding the usability of the proposed approach.

Thus, my vote to reject the paper remains.

**Limitations:**

The provided manuscript presents results only on a custom dataset, showing limited improvements compared to state-of-the-art methods like simple zero padding. Furthermore, the proposed method is much less efficient than zero padding, and the authors even suggest it might be ineffective for other datasets or model architectures. The outlined future improvements are valid and should be addressed in the next submission for this paper to be accepted at a top-tier conference. Therefore, I recommend rejecting the paper.

**Quality:**

1

**Strengths And Weaknesses:**

Strength:

The idea of using an autoregressive predictive model to improve padding alignment, in contrast to standard zero padding (which can add extraneous edges) or reflection padding (which might add incorrect content), is interesting.

Weaknesses:

1. The applicability of the proposed approach seems limited, as the authors themselves note. The dataset used for evaluation offers only restricted comparability to other state-of-the-art methods. Moreover, the authors' statement in line 192 "Our results might not directly apply to other CNN architectures and tasks" raises questions about this work's broader relevance to the community.
2. The computational overhead required to fit the model doesn't yield sufficient performance gains to justify its use in future work.
3. The authors use the abbreviation "RVSR" without introducing it properly.
4. Despite the encouraging future work section, this paper needs to incorporate the outlined improvements to be considered for a top-tier venue such as NeurIPS.

---

> ### Author Rebuttal · Authors · 2025-07-30
>
> Thank you for your review.
>
> ## Weaknesses
>
> > 1. [...] The dataset used for evaluation offers only restricted comparability to other state-of-the-art methods. Moreover, the authors' statement in line 192 "Our results might not directly apply to other CNN architectures and tasks" raises questions about this work's broader relevance to the community.
>
> We now have results on a segmentation task using several architectures, see our answer to reviewer dtmP's question 1. Generalizability.
>
> > 3. The authors use the abbreviation "RVSR" without introducing it properly.
>
> In the original work the method was called RVSR without an explanation as an abbreviation. We have correctly introduced RVSR as a convolutional super-resolution model.
>
> ## Questions
>
> > 1. Why are the predicted pixels in Figure 4 (top-left) not actually at the top, but rather in the middle of the feature map/image? It appears the proposed method predicts padding values that should typically be at the corners/edges of the feature map/image.
>
> The left part of Fig. 4 is about fitting the autoregressive model to the given image data, and the right part is about padding using the fitted model. We will rewrite the caption to make this more clear.
>
> > 2. In line 83, the authors state, "The approach is known as the covariance method." Could the authors please provide a reference for this claim?
>
> We will add a citation "(see Makhoul 1975 and Fig. 4.4 on page 69 of Weinlich 2022)".
>
> > 3. Why are some results in Table1 marked in gray?
>
> We wrote in the caption: "gray means an insufficient number of runs". We will clarify this, for example: "values written in gray are calculated from only 3 training runs and are therefore not statistically reliable".
>
> > 4. What do the different datapoints/shapes/colors represent in Figure 9? Please indicate by adding a legend or even improve visualization by using a different chart style like boxplots.
>
> The shapes represent different seeds used during training. We will add a legend (from Appendix F) disclosing the numerical seed values.

---

> > ### Comment · Reviewer_2k7F · 2025-08-01
> >
> > Thank you for the detailed rebuttal and for providing the additional analysis. While I appreciate the effort to clarify your position, my original concerns remain.
> >
> > I understand the argument that the segmentation experiments were meant to showcase the generalizability of the proposed approach. However, the results show that the proposed approach fails to achieve competitive results compared to much simpler approaches like zero padding. The performance gap and the line of argumentation made by the authors on which segmentation tasks can be considered suggests that the core approach is not robust enough to generalize effectively, which is precisely my concern.
> >
> > Further, the authors did not comment on the computational overhead of their proposed method, which is another major concern regarding the usability of the proposed approach.

---

### Official Review · Reviewer_g2mS · 2025-07-02

**Clarity:** 2
**Significance:** 2
**Originality:** 3
**Rating:** 3
**Confidence:** 3

**Summary:**

This paper proposes a padding method for convolutional neural networks, termed linear prediction padding (LP). LP employs stochastic autoregressive models to predict edge pixels by minimizing noise through least-squares fitting for each channel. It includes two variants: covariance-based and autocorrelation-based. The method has been validated in the context of satellite image super-resolution using the RVSR model. The LP reduces the mean squared error (MSE) by approximately 2% at the image edges compared to zero or replication padding, at the cost of a 25% increase in inference time.

**Questions:**

1.	Would the tiling issue mentioned in the paper occur in transformer-based networks? Can the proposed method be applied to transformer-based networks?
2.	Can the proposed method be effective on tasks other than satellite image super-resolution?
3.	Since the paper mainly compares MSE, can the proposed method be evaluated using other commonly used metrics, such as PSNR, SSIM, or LPIPS?
4.	The performance improvement is not significant, but the method has a impact on efficiency (~25%). Additionally, for tasks where edge quality is not critical, cropping the edges of the output nearly nullifies the advantage of the proposed method, though efficiency is still affected. Given that the advantages of the new method seem limited, is there any other potential significance or benefit?
5.	From Figure 1, it appears that the padding using the proposed linear prediction method still results in some blurriness, and it does not seem to provide much additional information compared to traditional padding methods. Can you explain why the proposed method still generates better results than zero padding and replicate padding, despite these observations?
6.	The paper mentions a related padding module, "multi-channel linear prediction," in line 29. However, it does not appear to be compared with the proposed method.

**Ethical Concerns:**

["NO or VERY MINOR ethics concerns only"]

**Final Justification:**

Thank you for the detailed response, which has addressed some of my concerns.

However, I remain unconvinced about the overall effectiveness of the proposed method. Both the paper and the response demonstrate limited improvements, which may not adequately substantiate the claimed contributions. Notably, other reviewers (e.g., Reviewers 2k7F, dtmP, Skv7) have expressed similar concerns.

As such, I will maintain my initial rating.

**Limitations:**

yes

**Paper Formatting Concerns:**

No major formatting issues were observed in this paper.

**Quality:**

3

**Strengths And Weaknesses:**

Strengths
1. The paper presents a complete derivation and proof of the process behind the proposed method, demonstrating its theoretical foundation and validity.
2. Conduct experiments from different aspects to comprehensively compare the performance of the proposed linear prediction padding with zero padding and replicate padding.

Weaknesses
1. Evaluated only on RVSR for satellite imagery; performance on other architectures (e.g., transformers) or data types (e.g., natural images) remains unverified.
2. The writing of the paper is a bit hard to understand for non-expert readers, and it lacks a related work section to provide a brief overview of the task and related methods.

---

> ### Author Rebuttal · Authors · 2025-07-30
>
> Thank you for your review.
>
> ## Weaknesses
>
> > 2. The writing of the paper is a bit hard to understand for non-expert readers, and it lacks a related work section to provide a brief overview of the task and related methods.
>
> We struggled with the 9-page limit and this affected our writing. Camera-ready has an extra page so we can write in less terse way the most difficult sentences/parts. We welcome input on which ones those are.
>
> ## Questions
> > 1. Would the tiling issue mentioned in the paper occur in transformer-based networks? Can the proposed method be applied to transformer-based networks?
> > 2. Can the proposed method be effective on tasks other than satellite image super-resolution?
>
> Non-overlapped tiled processing using transformer models also results in edge artifacts if not mitigated (see the middle column of Fig. 9 in Pérez et al. 2024). The mitigation suggested by the authors is not padding but overlapped patches.
>
> We have now experimented with different padding methods on a segmentation task using various encoders in a U-net, including a Mix Transformer encoder, see our answer to reviewer dtmP's question *1. Generalizability*. In general sequence padding use with transformers, attention masking would prevent attending to the padding and the padding method choice from affecting results.
>
> > 3. Since the paper mainly compares MSE, can the proposed method be evaluated using other commonly used metrics, such as PSNR, SSIM, or LPIPS?
>
> We used MSE, calculated over each shell to detect elevated error near image edges, and to match the MSE training loss for increased sensitivity. SSIM and LPIPS are not uniquely defined for evaluating on shells and would also be less appropriate error measures for the non-generative RVSR model and for possible downstream earth observation tasks that require exact geolocationing. We briefly discuss a potential generative extension of our padding method in Appendix B. PSNR is not uniquely defined in the tiled processing context. The need to normalize the error by using PSNR is reduced by that Sentinel 2 Level-1C satellite images have been normalized for top-of-atmosphere solar irradiance.
>
> > 4. [...] Given that the advantages of the new method seem limited, is there any other potential significance or benefit?
>
> We speculate that "larger effective receptive fields may favor `lp`" (lines 193-194). Perhaps there is a task--architecture combination for which this is true. We do not know yet. Publishing the method including code (now also PyTorch) enables others to test on their tasks.
>
> > 5. From Figure 1, it appears that the padding using the proposed linear prediction method still results in some blurriness, and it does not seem to provide much additional information compared to traditional padding methods. Can you explain why the proposed method still generates better results than zero padding and replicate padding, despite these observations?
>
> Our padding method predicts blurry expectances rather than generatively outpainting by sampling from a distribution of sharp images. We provide evidence (see Fig. 11) that linear prediction still extrapolates feature maps of the RVSR network with a smaller error than the other evaluated padding methods.
>
> > 6. The paper mentions a related padding module, "multi-channel linear prediction," in line 29. However, it does not appear to be compared with the proposed method.
>
> We also think that this is a limitation in our work and that it would be interesting to compare padding module to the other methods. However, padding module categorically differs from the presently evaluated methods in that it has trainable coefficients that are trained separately. Our evaluation would require significant modification to fairly include such methods. The additional hyperparameter search needed for padding module would significantly increase the evaluation computational budged. We only recently identified padding module as a form of linear prediction (a term not mentioned in the original work) and included the reference for completeness.
>
> ## References
>
> Pérez, Antonio, et al. "Transformer based super-resolution downscaling for regional reanalysis: Full domain vs tiling approaches." arXiv preprint arXiv:2410.12728 (2024).

---

> > ### Comment · Reviewer_g2mS · 2025-08-06
> >
> > Thank you for the detailed response, which has addressed some of my concerns.
> >
> > However, I remain unconvinced about the overall effectiveness of the proposed method. Both the paper and the response demonstrate limited improvements, which may not adequately substantiate the claimed contributions. Notably, other reviewers (e.g., Reviewers 2k7F, dtmP, Skv7) have expressed similar concerns.
> >
> > As such, I will maintain my initial rating.

---

### Official Review · Reviewer_Skv7 · 2025-07-02

**Clarity:** 2
**Significance:** 2
**Originality:** 2
**Rating:** 2
**Confidence:** 3

**Summary:**

The paper proposes a per-channel, differentiable padding method for convolutional neural networks (CNNs) called linear prediction (lp) padding, which adapts autoregressive linear models to predict input tiles beyond their boundaries. The approach is parameter-free and can be implemented in a differentiable framework. The authors apply linear prediction padding in satellite image super-resolution using the RVSR model and compare it with standard zero, replication, and extrapolation padding methods. Experiments indicate that lp padding reduces the mean square error (especially near image edges) compared to baselines, although at the cost of a moderately increased inference time. The paper includes theoretical justifications, practical considerations, and ablation across cropping schemes, and discusses the implications of padding choice for CNN capacity usage.

**Questions:**

See the weaknesses.

**Ethical Concerns:**

["NO or VERY MINOR ethics concerns only"]

**Final Justification:**

My concerns about the computational complexity and limited performance are not well addressed in this paper. Thus, I keep my original score to reject this paper.

**Limitations:**

Yes

**Quality:**

2

**Strengths And Weaknesses:**

Strengths:
-This is conceptually clear, does not require additional network parameters.
-The authors carefully compare multiple padding strategies (lp variants, zero, repl, extrapolation), with experiments using a modern.

Weaknesses:
-The reported improvement in mean test MSE is modest (e.g., 0.2–0.9% overall and 1.5–2% at the outermost shell compared to standard padding, per Table 1). For many applications, this benefit may not outweigh the ~25% increased inference cost.
-While implementation is differentiable and parameter-free, inference cost rises the method trades speed/memory efficiency for small quality increases, which may hinder adoption for large-scale or real-time systems.
-Multiple lp variants are introduced (based on different neighborhoods; covariance vs. autocorrelation computation), which, while comprehensive, may make practical adoption cumbersome. It is not immediately clear to users which variant offers the best tradeoff for a new use case.
-Although the motivation is strong for satellite imagery tiling, the experimental scope is limited to a single image domain and model (RVSR). There is no empirical demonstration on other vision tasks (e.g., segmentation, classification) or with alternative architectures (such as U-Net, ResNet).
-While Table 1 is detailed, it is dense—reporting many variants, metrics and intervals—making it difficult to quickly identify the practical significance or to directly answer which method should generally be preferred.

---

> ### Author Rebuttal · Authors · 2025-07-30
>
> Thank you for your review.
>
> ## Weaknesses
>
> > -Multiple lp variants are introduced (based on different neighborhoods; covariance vs. autocorrelation computation), which, while comprehensive, may make practical adoption cumbersome. It is not immediately clear to users which variant offers the best tradeoff for a new use case.
> > -While Table 1 is detailed, it is dense—reporting many variants, metrics and intervals—making it difficult to quickly identify the practical significance or to directly answer which method should generally be preferred.
>
> We will add a recommendation to try `lp1x1cs` for new use cases based on that it is the simplest and fastest variant, the easiest to implement, and likely to give most of the benefits of heavier variants. It can also be used for multi-dimensional data in addition to 2D data.
>
> > -Although the motivation is strong for satellite imagery tiling, the experimental scope is limited to a single image domain and model (RVSR). There is no empirical demonstration on other vision tasks (e.g., segmentation, classification) or with alternative architectures (such as U-Net, ResNet).
>
> We now have results on a segmentation task using several architectures, see our answer to reviewer dtmP's question 1. Generalizability.

---

> > ### Comment · Reviewer_Skv7 · 2025-08-06
> >
> > Thank you for your response. The rebuttal only addresses some concerns. My concerns about the trade-off between computation and performance, and generalization still remain in this feedback. In particular, the authors directly neglect my comment about the limited performance improvement. Moreover, the reported segmentation of the proposed method in this rebuttal cannot significantly outperform the baselines.

---

### Official Review · Reviewer_dtmP · 2025-07-02

**Clarity:** 2
**Significance:** 2
**Originality:** 2
**Rating:** 3
**Confidence:** 3

**Summary:**

This paper introduces linear prediction (LP) padding, a novel non-trainable and differentiable padding technique for convolutional neural networks processing 2D spatial data. The core innovation lies in its per-channel autoregressive modeling approach:

1. Methodology: For each channel, a stochastic autoregressive linear model is fitted to the data by minimizing noise terms via least-squares optimization. Padding values are iteratively generated as conditional expectations from this model.
2. Validation: Applied to the RVSR super-resolution model trained on satellite imagery, the simplest LP variant achieves:
   - ∼2% reduction in mean square error (MSE) at image edges compared to zero/replication padding
   - ∼25% increase in inference latency
3. Key observations:
   - LP provides better approximations of both input data and feature maps
   - Zero padding forces the model to dedicate more capacity to compensate for edge errors
   - Output cropping minimizes padding-induced artifacts and equalizes performance across methods

This work establishes LP padding as a mathematically grounded alternative to conventional padding methods, with demonstrated efficacy in edge-quality enhancement for super-resolution tasks.

**Questions:**

1. Generalizability: Can LP padding improve performance in non-satellite tasks (e.g., natural images/medical data) or other architectures (e.g., U-Nets, Transformers)? Add experiments on 1–2 diverse datasets/architectures.
2. Efficiency-Scalability Trade-off: LP adds 25% inference latency. Can optimizations (e.g., precomputed coefficients) mitigate this? How does cost scale for larger neighborhoods (e.g., lp6x7)? Report latency/FLOPs for all LP variants.
3. Fig 1 Utility: What is the purpose of Fig 1? It is never cited. Integrate Fig 1 into the narrative or remove it.

**Ethical Concerns:**

["NO or VERY MINOR ethics concerns only"]

**Limitations:**

Yes.

**Paper Formatting Concerns:**

Fig 1 not referenced: Violates NeurIPS formatting (all figures must be cited). Add in-text reference or remove.

**Quality:**

2

**Strengths And Weaknesses:**

Strengths:

- Quality: Technically sound with rigorous implementation (JAX RVSR reimplementation, ablation via output cropping).
- Originality: Novel application of autoregressive models to padding, offering a parameter-free alternative.
- Clarity: Methods well-structured (Sections 2–3); figures (e.g., Fig 6–7) clarify cropping/padding workflows.
- Significance: Highlights CNN capacity trade-offs when compensating padding errors.

Weaknesses:

- Significance: Marginal gains at high cost. Limited motivation for real-world adoption.
- Quality: Experiments only on model (RVSR) and task (satellite SR). Generalizability unproven.
- Clarity: Fig 1 unused (no reference in text); covariance variants underperformed but lacked analysis.
- Originality: LP is an established signal-processing technique; innovation lies in CNN adaptation but not transformative.

---

> ### Author Rebuttal · Authors · 2025-07-30
>
> Thank you for your review.
>
> ## Questions
> > 1. Generalizability: Can LP padding improve performance in non-satellite tasks (e.g., natural images/medical data) or other architectures (e.g., U-Nets, Transformers)? Add experiments on 1–2 diverse datasets/architectures.
>
> We have now used the 670-image 2D light microscopy nuclei **binary segmentation** dataset CellNucleiMSBench from MedSegBench (Aydin et al. 2024) to evaluate the padding method `lp1x1cs`, `repl`, and `zero`, by substituting the padding method in the 2D convolutions of U-nets that use different encoder architectures. Of the MedSegBench datasets, this dataset best resembled a tiled processing task. We used the 200-epoch training code and the evaluation code from MedSegBench, and an unoptimized PyTorch implementation of our padding method which we will also publish with an MIT license.
>
> The results below, obtained with 9 seeds, indicate that for this task and training setup the Mix Transformer MiT-B0 (Xie et al. 2021) encoder which contains padded convolutions and the Mobile-Net-v2 encoder favor `zero` padding when evaluated using the F1 metric which is directly related to the Dice training loss used. For the remaining convolutional encoders any differences due to padding methods appear to be dwarfed by training stochasticity. We will increase the number of seeds to 20 or so for better statistics.
>
> Mean test set scores (higher is better for all scores, highest for each model is highlighted) ±2 standard error of the mean:
> |Model|Padding|F1|PREC|REC|ACC|IOU|
> |-|-|-|-|-|-|-|
> |ResNet-18|`lp1x1cs`|0.27 ±0.006|**0.244 ±0.008**|0.378 ±0.018|**0.966 ±0.002**|0.163 ±0.004
> ||`repl`|0.261 ±0.003|0.228 ±0.01|0.403 ±0.022|0.962 ±0.003|0.156 ±0.002
> ||`zero`|**0.28 ±0.004**|0.237 ±0.009|**0.42 ±0.022**|0.964 ±0.002|**0.168 ±0.003**
> |ResNet-50|`lp1x1cs`|0.268 ±0.005|0.243 ±0.007|0.364 ±0.018|**0.967 ±0.002**|0.162 ±0.003
> ||`repl`|0.263 ±0.005|**0.249 ±0.012**|0.36 ±0.025|0.967 ±0.002|0.158 ±0.004
> ||`zero`|**0.273 ±0.003**|0.238 ±0.007|**0.414 ±0.017**|0.962 ±0.003|**0.164 ±0.002**
> |Efficient-Net|`lp1x1cs`|**0.237 ±0.008**|**0.218 ±0.007**|0.336 ±0.036|0.966 ±0.003|**0.14 ±0.006**
> ||`repl`|0.209 ±0.007|0.213 ±0.009|0.267 ±0.031|**0.967 ±0.003**|0.121 ±0.005
> ||`zero`|0.237 ±0.009|0.21 ±0.007|**0.347 ±0.027**|0.963 ±0.002|0.139 ±0.006
> |Mobile-Net-v2|`lp1x1cs`|0.198 ±0.003|0.185 ±0.008|0.286 ±0.017|0.962 ±0.002|0.113 ±0.002
> ||`repl`|0.19 ±0.009|0.18 ±0.008|**0.302 ±0.024**|0.957 ±0.004|0.109 ±0.005
> ||`zero`|**0.206 ±0.005**|**0.209 ±0.009**|0.27 ±0.015|**0.965 ±0.003**|**0.119 ±0.003**
> |MiT-B0|`lp1x1cs`|0.241 ±0.002|0.183 ±0.005|0.479 ±0.019|0.949 ±0.003|0.145 ±0.001
> ||`repl`|0.241 ±0.003|0.174 ±0.005|**0.509 ±0.018**|0.945 ±0.003|0.144 ±0.002
> ||`zero`|**0.258 ±0.003**|**0.195 ±0.007**|0.485 ±0.023|**0.953 ±0.003**|**0.154 ±0.002**
>
> The same results as differences to the `zero` baseline:
> |Model|Padding|F1 diff|PREC diff|REC diff|ACC diff|IOU diff|
> |-|-|-|-|-|-|-|
> |ResNet-18|`lp1x1cs`|-0.01 ±0.008|**0.007 ±0.014**|-0.043 ±0.027|**0.002 ±0.003**|-0.005 ±0.006
> ||`repl`|-0.018 ±0.007|-0.009 ±0.013|-0.017 ±0.034|-0.002 ±0.003|-0.013 ±0.005
> ||`zero`|**0.0 ±0.0**|0.0 ±0.0|**0.0 ±0.0**|0.0 ±0.0|**0.0 ±0.0**
> |ResNet-50|`lp1x1cs`|-0.005 ±0.006|0.005 ±0.007|-0.051 ±0.019|**0.005 ±0.002**|-0.002 ±0.004
> ||`repl`|-0.01 ±0.007|**0.011 ±0.016**|-0.054 ±0.034|0.005 ±0.004|-0.006 ±0.005
> ||`zero`|**0.0 ±0.0**|0.0 ±0.0|**0.0 ±0.0**|0.0 ±0.0|**0.0 ±0.0**
> |Efficient-Net|`lp1x1cs`|**0.0 ±0.01**|**0.008 ±0.012**|-0.011 ±0.045|0.002 ±0.004|**0.001 ±0.007**
> ||`repl`|-0.028 ±0.008|0.003 ±0.014|-0.08 ±0.047|**0.004 ±0.005**|-0.019 ±0.005
> ||`zero`|0.0 ±0.0|0.0 ±0.0|**0.0 ±0.0**|0.0 ±0.0|0.0 ±0.0
> |Mobile-Net-v2|`lp1x1cs`|-0.009 ±0.006|-0.024 ±0.007|0.015 ±0.019|-0.003 ±0.003|-0.006 ±0.004
> ||`repl`|-0.016 ±0.012|-0.029 ±0.013|**0.032 ±0.029**|-0.008 ±0.004|-0.01 ±0.007
> ||`zero`|**0.0 ±0.0**|**0.0 ±0.0**|0.0 ±0.0|**0.0 ±0.0**|**0.0 ±0.0**
> |MiT-B0|`lp1x1cs`|-0.017 ±0.004|-0.012 ±0.006|-0.005 ±0.024|-0.004 ±0.003|-0.01 ±0.002
> ||`repl`|-0.017 ±0.005|-0.021 ±0.008|**0.024 ±0.027**|-0.008 ±0.004|-0.01 ±0.003
> ||`zero`|**0.0 ±0.0**|**0.0 ±0.0**|0.0 ±0.0|**0.0 ±0.0**|**0.0 ±0.0**
>
> The results are for whole images. Due to a large class imbalance, we were unable to evaluate performance on shells which rarely see nuclei.
>
> We did initially try to find an open-license segmentation dataset but were unable to find any without built-in edge artifacts which we think would have artificially reduced the usability of our edge-hiding padding method intended to be used in the context of tiled processing of larger images. Therefore we went with the tiled processing image-to-image super-resolution task. CellNucleiMSBench is likelty to contain edge artifacts as the manual segmentation has likely been done on the spatial extent of the dataset images rather than on larger images from which the dataset would be cropped. Also resizing operations by MedSegBench and upstream work to the final resolution of 512x512 creates edge artifacts. Because of these intricacies and due to the small size of the dataset we will include the new results as supplementary, summarizing the results in main text without making a final conclusion on generalizability. We will note that based on the results, our padding method is unlikely to improve results of non-tiled processing. It would be possible to built a custom segmentation dataset that does not contain edge artifacts, for example by cropping land cover data, but for the time being we lack the resources to do that and to evaluate on a new task.
>
> > 2. Efficiency-Scalability Trade-off: LP adds 25% inference latency. Can optimizations (e.g., precomputed coefficients) mitigate this? How does cost scale for larger neighborhoods (e.g., lp6x7)? Report latency/FLOPs for all LP variants.
>
> We have optimized our current method which as defined computes coefficients based on the current input. In Future work, we have suggested trying batch/historical covariance statistic accumulation. This would use precomputed coefficients in inference but also give different results than the current method, so not a pure optimization without side-effects. We have reported inference throughput for all `lp` variants in Table 1 and Fig. 10. We will improve on how those results are summarized in text and also clarify that the covariance-method-based variants `lp1x1cs` and `lp2x1cs` use smaller 1D neighborhoods than the 2D-neighborhood autocorrelation-based variants and thus only capture simple (but largely sufficient) 1D dependencies in the data.
>
> > 3. Fig 1 Utility: What is the purpose of Fig 1? It is never cited. Integrate Fig 1 into the narrative or remove it.
>
> Apologies for the missing reference. We want to illustrate the method on the first page for the benefit of a casual reader. We will add the reference to the last paragraph of the introduction.
>
> ## References
>
> Aydin, Musa, and Zeki Kuş. "Medsegbench: A comprehensive benchmark for medical image segmentation in diverse data modalities." medRxiv (2024): 2024-08.
>
> Xie, Enze, et al. "SegFormer: Simple and efficient design for semantic segmentation with transformers." Advances in neural information processing systems 34 (2021): 12077-12090.

---

> > ### Author Response · Authors · 2025-08-05
> > **Updated segmentation task results with 30 seeds**
> >
> > As promosed, below are updated segmentation task results, now with a total of 30 seeds. The results indicate that for this task and training setup the Mix Transformer MiT-B0 encoder and with less certainty the ResNet-50 encoder favor `zero` padding when evaluated using the mean F1 score. For all the other encoders, `lp1x1cs` gives a higher mean F1 score than `repl` and the differences between `lp1x1cs` and `zero` are marginal and inconclusive.
> >
> > Mean test set scores (higher is better for all scores, highest for each model is highlighted) ±2 standard error of the mean:
> >
> > |Model|Padding|F1|PREC|REC|ACC|IOU|
> > |-|-|-|-|-|-|-|
> > |ResNet-18|`lp1x1cs`|0.275 ±0.003|**0.244 ±0.004**|**0.402 ±0.012**|0.964 ±0.001|0.165 ±0.002
> > ||`repl`|0.263 ±0.002|0.241 ±0.005|0.382 ±0.013|0.964 ±0.002|0.157 ±0.002
> > ||`zero`|**0.276 ±0.003**|0.242 ±0.005|0.393 ±0.011|**0.966 ±0.001**|**0.166 ±0.002**
> > |ResNet-50|`lp1x1cs`|0.262 ±0.003|0.234 ±0.004|0.375 ±0.013|**0.965 ±0.001**|0.157 ±0.002
> > ||`repl`|0.26 ±0.002|0.238 ±0.006|0.371 ±0.012|0.964 ±0.001|0.156 ±0.002
> > ||`zero`|**0.268 ±0.002**|**0.24 ±0.006**|**0.398 ±0.013**|0.963 ±0.002|**0.16 ±0.001**
> > |Efficient-Net|`lp1x1cs`|0.236 ±0.004|**0.223 ±0.004**|0.316 ±0.017|0.967 ±0.001|0.139 ±0.003
> > ||`repl`|0.212 ±0.003|0.215 ±0.004|0.261 ±0.012|**0.968 ±0.001**|0.122 ±0.002
> > ||`zero`|**0.238 ±0.005**|0.216 ±0.004|**0.325 ±0.013**|0.966 ±0.001|**0.14 ±0.003**
> > |Mobile-Net-v2|`lp1x1cs`|0.2 ±0.003|0.2 ±0.004|0.27 ±0.009|**0.965 ±0.001**|0.115 ±0.002
> > ||`repl`|0.188 ±0.006|0.19 ±0.005|0.274 ±0.014|0.962 ±0.002|0.108 ±0.004
> > ||`zero`|**0.204 ±0.003**|**0.2 ±0.005**|**0.287 ±0.012**|0.963 ±0.002|**0.118 ±0.002**
> > |MiT-B0|`lp1x1cs`|0.241 ±0.002|0.181 ±0.003|0.478 ±0.01|0.948 ±0.002|0.145 ±0.001
> > ||`repl`|0.24 ±0.002|0.174 ±0.003|**0.497 ±0.011**|0.946 ±0.002|0.144 ±0.001
> > ||`zero`|**0.257 ±0.002**|**0.196 ±0.003**|0.471 ±0.011|**0.954 ±0.001**|**0.154 ±0.001**
> >
> > The same results as differences to the zero baseline:
> >
> > |Model|Padding|F1 diff|PREC diff|REC diff|ACC diff|IOU diff|
> > |-|-|-|-|-|-|-|
> > |ResNet-18|`lp1x1cs`|-0.002 ±0.004|**0.002 ±0.007**|**0.009 ±0.015**|-0.001 ±0.002|-0.0 ±0.003
> > ||`repl`|-0.013 ±0.004|-0.001 ±0.008|-0.011 ±0.02|-0.001 ±0.002|-0.008 ±0.003
> > ||`zero`|**0.0 ±0.0**|0.0 ±0.0|0.0 ±0.0|**0.0 ±0.0**|**0.0 ±0.0**
> > |ResNet-50|`lp1x1cs`|-0.006 ±0.004|-0.006 ±0.007|-0.023 ±0.019|**0.002 ±0.002**|-0.003 ±0.002
> > ||`repl`|-0.008 ±0.003|-0.002 ±0.009|-0.027 ±0.019|0.001 ±0.002|-0.005 ±0.002
> > ||`zero`|**0.0 ±0.0**|**0.0 ±0.0**|**0.0 ±0.0**|0.0 ±0.0|**0.0 ±0.0**
> > |Efficient-Net|`lp1x1cs`|-0.002 ±0.005|**0.007 ±0.006**|-0.009 ±0.019|0.002 ±0.002|-0.001 ±0.004
> > ||`repl`|-0.026 ±0.005|-0.001 ±0.007|-0.064 ±0.017|**0.003 ±0.002**|-0.018 ±0.004
> > ||`zero`|**0.0 ±0.0**|0.0 ±0.0|**0.0 ±0.0**|0.0 ±0.0|**0.0 ±0.0**
> > |Mobile-Net-v2|`lp1x1cs`|-0.004 ±0.005|-0.0 ±0.006|-0.018 ±0.016|**0.002 ±0.002**|-0.003 ±0.003
> > ||`repl`|-0.017 ±0.007|-0.01 ±0.006|-0.013 ±0.018|-0.002 ±0.002|-0.01 ±0.004
> > ||`zero`|**0.0 ±0.0**|**0.0 ±0.0**|**0.0 ±0.0**|0.0 ±0.0|**0.0 ±0.0**
> > |MiT-B0|`lp1x1cs`|-0.016 ±0.002|-0.015 ±0.004|0.007 ±0.013|-0.006 ±0.002|-0.009 ±0.002
> > ||`repl`|-0.017 ±0.003|-0.022 ±0.004|**0.026 ±0.013**|-0.008 ±0.002|-0.01 ±0.002
> > ||`zero`|**0.0 ±0.0**|**0.0 ±0.0**|0.0 ±0.0|**0.0 ±0.0**|**0.0 ±0.0**

---

### Note · Authors · 2025-08-12

We thank the reviewers for their additional comments. We agree with the concerns regarding undemonstrated generalizability and the relatively small error reduction compared to the throughput decrease, which are already discussed in the paper. To improve clarity, we will add an explicit note stating that generalizability to other tasks and models remains to be demonstrated.

---

### Decision · Program_Chairs · 2025-09-17

**Decision:**

Reject

**Comment:**

The paper proposes a new method and provides experiments to demonstrate its generalizability across different tasks. While the problem addressed is relevant, the overall evaluation leaves significant doubts about the effectiveness and robustness of the proposed approach.

Multiple reviewers, share concerns that the improvements shown in the paper are limited and do not sufficiently substantiate the claimed contributions. In particular, the segmentation experiments, intended to demonstrate generalizability, fail to achieve competitive performance - even underperforming compared to much simpler baselines such as zero padding. This performance gap directly undermines the paper’s central claim of robustness and general utility.

Moreover, important practical aspects remain unaddressed. Specifically, the computational overhead of the method is not analyzed or discussed, raising further concerns about its usability in real-world scenarios. While the rebuttal provided clarifications, it did not convincingly resolve these core issues.